# PartCrafter: Structured 3D Mesh Generation via Compositional Latent Diffusion Transformers

**Yuchen Lin**[1,3][*], **Chenguo Lin**[1][*], **Panwang Pan**[2][†],
**Honglei Yan**[2], **Yiqiang Feng**[2], **Yadong Mu**[1], **Katerina Fragkiadaki**[3]
[*] Equal contribution [†] Project lead
[1]Peking University, [2]ByteDance, [3]Carnegie Mellon University
https://wgsxm.github.io/projects/partcrafter

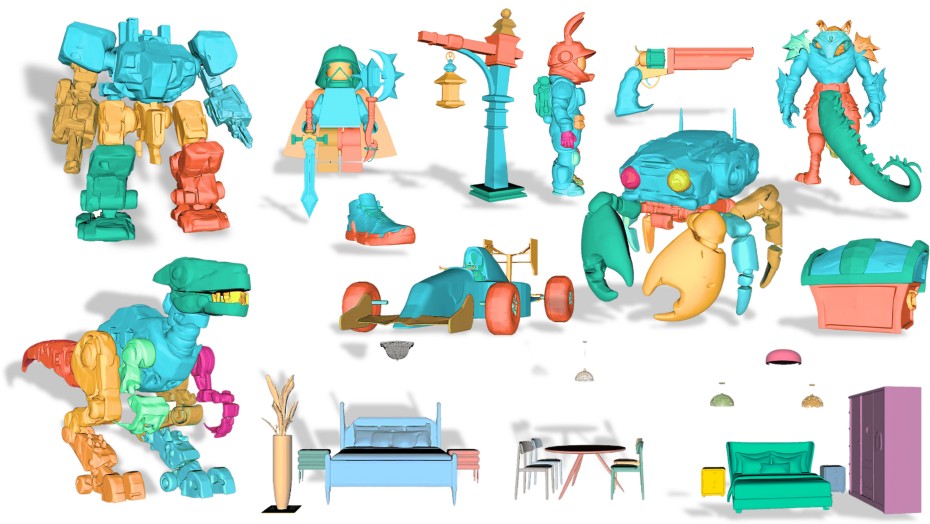

Figure 1: **PARTCRAFTER** is a structured 3D generative model that **jointly generates multiple parts and objects from a single RGB image in one shot**, without the need for segmented image inputs.

## Abstract

We introduce PARTCRAFTER, the first structured 3D generative model that jointly synthesizes multiple semantically meaningful and geometrically distinct 3D meshes from a single RGB image. Unlike existing methods that either produce monolithic 3D shapes or follow two-stage pipelines, *i.e.* first segmenting an image and then reconstructing each segment, PARTCRAFTER adopts a unified, compositional generation architecture that does not rely on pre-segmented inputs. Conditioned on a single image, it simultaneously denoises multiple 3D parts, enabling end-to-end part-aware generation of both individual objects and complex multi-object scenes. PARTCRAFTER builds upon a pretrained 3D mesh diffusion transformer (DiT) trained on whole objects, inheriting the pretrained weights, encoder, and decoder, and introduces two key innovations: (1) **A compositional latent space**, where each 3D part is represented by a set of disentangled latent tokens; (2) **A hierarchical attention mechanism** that enables structured information flow both within individual parts and across all parts, ensuring global coherence while preserving part-level detail during generation. To support part-level supervision, we curate a new dataset by mining part-level annotations from large-scale 3D object datasets. Experiments show that PARTCRAFTER outperforms existing approaches in generating decomposable 3D meshes, including parts that are not directly visible in input images, demonstrating the strength of part-aware generative priors for 3D understanding and synthesis. Code and training data are released.

39th Conference on Neural Information Processing Systems (NeurIPS 2025).

# 1  Introduction

A central organizing principle in perception is the role of objects and parts—semantically coherent units that serve as compositional building blocks for higher-level cognitive tasks, including language, planning, and reasoning. This part-based structure facilitates generalization, as parts can be independently interpreted, recombined, and reused across different contexts. In contrast, most contemporary neural networks lack the ability to form and manipulate such structured, symbol-like entities.

In 3D generation, diffusion-based generative models have shown strong capabilities in synthesizing entire 3D object meshes from scratch or from images [1, 2, 3]. However, these models typically operate at the whole-object level and do not support part-level decomposition. This limitation restricts their applicability in downstream tasks such as texture mapping, animation, physical simulation, and scene editing. Some recent efforts have attempted to address this by first segmenting images into semantic parts and then reconstructing each part in 3D [4, 5, 6, 7]. However, this two-stage pipeline suffers from errors in segmentation, extensive computational costs for extra segmentation models, and difficulties in scaling up, thereby limiting both robustness and fidelity.

We introduce PARTCRAFTER, a structured generative model for 3D scenes that enables part-level generation from a single RGB image through a compositional latent space. PARTCRAFTER jointly generates multiple distinct 3D parts by binding each to a dedicated set of latent variables. This disentanglement allows parts to be independently edited, removed, or added without disrupting the rest of the scene. PARTCRAFTER builds upon large-scale pretrained 3D object latent diffusion models, which represent object meshes as sets of latent tokens [8] aligned either explicitly [3] or implicitly [1, 2] to regions in 3D space. PARTCRAFTER restructures these pretrained models into a compositional architecture equipped with a varying number of latent token sets. It guides each latent token set during the denoising process to associate with a particular 3D part entity, through a novel local-global attention mechanism that facilitates both intra-part and inter-part information flow, in an identity-aware and permutation-invariant way. A shared decoder then maps each latent set into a coherent 3D mesh. We initialize the model's encoder, decoder, and denoising transformer using weights pretrained on whole-object mesh generation tasks [1]. When conditioned on a single RGB image, the model produces structured 3D outputs with coherent part-level decomposition, eliminating the need for brittle segmentation-then-reconstruction pipelines.

To support training, we construct a large-scale dataset by mining part-level annotations from existing 3D object repositories. Many assets in datasets like Objaverse [9] contain part information in their GLTF metadata, as they are often authored using modular components. Rather than flattening these into single meshes, as done in prior work, we retain their part annotations. Our curated dataset merges Objaverse [9], ShapeNet [10], and the Amazon Berkeley Objects (ABO) dataset [11], resulting in a rich collection of part-annotated 3D models suitable for learning compositional generation. As for scene-level generation, we leverage the existing 3D scene dataset 3D-Front [12] for training.

We evaluate PARTCRAFTER on both 3D part-level object generation and 3D scene reconstruction, and compare it against existing two-stage methods [6, 7] that first segment the input image and then reconstruct each segment. Our results show that PARTCRAFTER achieves higher generation quality and better efficiency. As shown in our experimental section, PARTCRAFTER can automatically infer invisible 3D structures from a single image. It can be equally well used at the object or scene level, which makes it a universal model for 3D scene reconstruction. Notably, PARTCRAFTER surpasses its underlying 3D object generative model on object reconstruction fidelity, showing that understanding the compositional structure of objects enhances the quality of 3D generation.

In summary, our contributions are as follows:

- We propose PARTCRAFTER, a structured 3D generation model with explicit part-level binding, capable of generating semantically meaningful 3D components to compose an object or a scene from image prompts, without any segmentation input.
- We design a novel compositional DiT architecture with structured latent spaces, identity-aware part-global attention, and shared decoders.
- A new dataset with part annotations is curated from existing 3D assets.
- PARTCRAFTER achieves state-of-the-art performance on structured 3D generation tasks, demonstrating strong results on both individual objects and complex multi-object scenes. Comprehensive ablation studies validate the contribution of each component in our approach.

## 2 Related Work

**3D Object Generation**    Previous works on 3D object generation have adopted various representations, including voxels [13, 14, 15], point clouds [16, 17, 18], signed distance fields (SDFs) [19, 20], neural radiance fields (NeRFs) [21, 22, 23, 24, 25], and 3D Gaussian splitting (3DGS) [26, 27, 28]. We focus on 3D meshes for their compatibility with real-world 3D content creation pipelines. One line of mesh generation works produces explicit mesh structures autoregressively [29, 30, 31, 32, 33, 34]. The other line of 3D mesh generation methods uses latent diffusion models [3, 8, 35, 36, 37], and has demonstrated strong performance, particularly when training high-capacity diffusion models across large-scale datasets [1, 2, 38, 39, 40]. PARTCRAFTER builds upon the latter line of 3D mesh generative models. However, these methods typically generate 3D objects as holistic entities, neglecting the natural part-based decomposition that exists in 3D object datasets and is commonly employed by human artists during creation. PARTCRAFTER addresses this limitation.

**3D Part-level Object Generation**    3D part-level object generation is a long-standing problem in computer vision. One line of work [41, 42] has focused on assembling existing parts by inferring the right translations and orientations. Another line of work, which is more related to our work, generates 3D geometry of parts and their structure to form a complete object. Different primitives have been used as representation, such as point clouds [43], voxels [44], implicit neural fields [45, 46, 47, 48], and other parametric representations [49, 50, 51]. These works are limited by either the scale of datasets [10, 52] or the expressiveness of the representation. To improve the visual quality and utilize prior knowledge of 2D pretrained models, several recent works [4, 5] propose to leverage multi-view diffusion and 2D segmentation models [53] to generate NeRF [21] or NeuS [54] with part labels. Concurrent work HoloPart [6] proposes to first segment the 3D object mesh into parts and then refine the parts' geometry with a pretrained 3D DiT [1]. Although effective, these methods are heavily dependent on the segmentation quality and are difficult to scale up. In contrast, our method can generate structured 3D part-level objects in one shot without any 2D or 3D segmentation.

**3D Scene Generation**    Prior work on 3D object-composed scene generation typically extracts abstract representations such as layouts [55, 56], graphs [57, 58], or segmentations [59, 60, 61, 62, 63, 64] to model object relationships, and then generates [65, 66, 67] or retrieves [68, 68] objects based on these representations and input conditions. The closest work to ours is MIDI [7], which generates compositional 3D scenes by segmenting the input image [69] and prompting object-level 3D diffusion models with region segments. However, MIDI requires all components to be visible in the image, hindering its ability to segment and reconstruct occluded parts, which is a shared limitation across all two-stage segment-and-reconstruct methods. PARTCRAFTER is inspired by MIDI but overcomes this limitation. Without any additional segmentation model, PARTCRAFTER is capable of generating parts even when they are not visible in the conditioning image prompt. PARTCRAFTER demonstrates that explicit segmentation is not a necessary prerequisite for structured 3D scene generation.

## 3 PARTCRAFTER: A Compositional 3D Diffusion Model

The architecture of PARTCRAFTER is illustrated in Figure 2. It is a compositional generative model that generates structured 3D assets by simultaneously denoising several part-specified latent token sets. Specifically, given a prompt RGB image $\mathbf{c}$ and a user-specified number of parts $N$, PARTCRAFTER generates a structured 3D asset $\mathcal{O} := \{\mathbf{p}_i\}_{i=1}^N$, *e.g.*, a part-decomposable object or a 3D scene composed of object parts, where each part $\mathbf{p}_i$ is a separate, semantically meaningful component of the asset. 3D meshes are utilized as the output representation. At inference time, all part meshes $\mathbf{p}_i := \{\mathcal{V}_i, \mathcal{F}_i\}$ are generated and decoded simultaneously, where $\mathcal{V}_i \in \mathbb{R}^{V_i \times 3}$ is the vertex set and $\mathcal{F}_i \in \mathbb{N}^{F_i \times 3}$ is the face set of the $i$-th part. Notably, coordinates of each part's vertices are in the same global canonical space $[-1, 1]^3$, allowing for easy assembly of the parts into a complete object or scene, without any need for additional transformations.

PARTCRAFTER builds upon a pretrained 3D object mesh generation model [1] by utilizing and reassembling its encoders, decoders, and DiT blocks into a compositional multi-entity generation architecture. We provide a description of object-level 3D generation models in Section 3.1, followed by a detailed presentation of our proposed structured multi-entity model in Section 3.2.

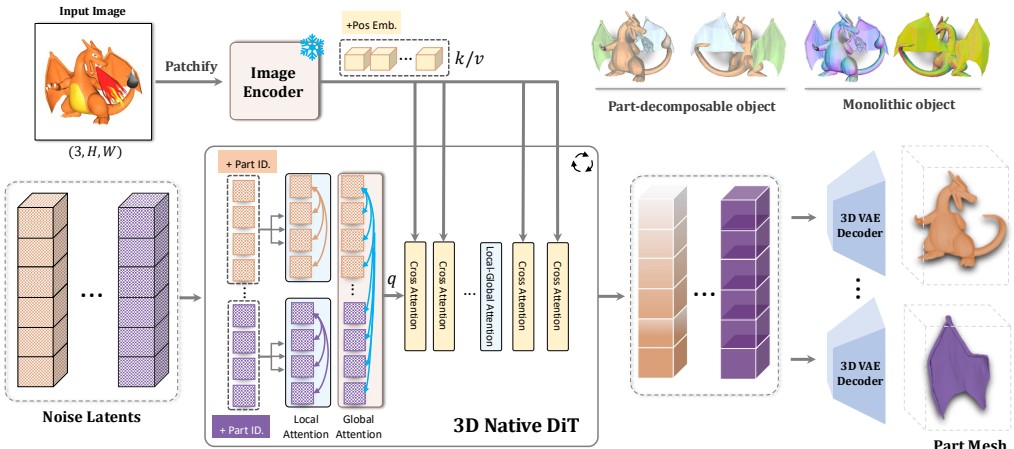

Figure 2: **Architecture of PARTCRAFTER.** Our model utilizes local-global attention to capture both part-level and global features. Part ID embeddings and incorporation of the image condition into both local and global features ensure the independence and semantic coherence of the generated parts.

## 3.1 Preliminary: Diffusion Transformer for 3D Object Mesh Generation

Pretrained 3D object generation models have shown impressive performance in generating high-quality 3D objects. Our model specifically builds on top of TripoSG [1], a state-of-the-art 3D image-to-mesh generative model. TripoSG first encodes 3D shapes into a set of latent vectors with a transformer-based Variational Autoencoder (VAE) using 3DShape2VecSet [8]. The decoder of the VAE uses a Signed Distance Function (SDF) representation, which enables sharper geometry and avoids aliasing. A rectified flow model is trained on the VAE latents to generate new 3D shapes from noise. The model is conditioned on a single input image using DINOv2 [70] features, injected via cross-attention in every transformer block. It is trained on 2 million high-quality 3D shapes curated from Objaverse [9, 71] and ShapeNet [10] through a rigorous four-stage pipeline involving quality scoring, filtering out noisy data, mesh repair, and conversion to a watertight SDF-compatible format.

## 3.2 PARTCRAFTER

The design of PARTCRAFTER is motivated by our observation that foundation 3D latent diffusion models such as TripoSG [1], although trained on object-level training examples, can be applied effectively out-of-the-box to auto-encode 3D parts as well, without normalizing the input 3D mesh to the center of the canonical space (as done for 3D object meshes). We thus adapt the neural components of TripoSG to build a part-decomposable 3D asset reconstruction model.

**Compositional Latent Space** The key insight of 3D object mesh generative models with a set of latents is that each token in the latent space of the pretrained 3D VAE [1, 2, 8] is implicitly associated with an area of the canonical 3D space. PARTCRAFTER expands the latent space of monolithic object generative models with multiple sets of latents, each taking care of a separate 3D part. Specifically, an object or a scene is represented as a set of $N$ parts $\mathcal{O} = \{\mathbf{p}_i\}_{i=1}^N$ and each part $\mathbf{p}_i$ is represented by a set of latent tokens $\boldsymbol{z}_i = \{z_{ij}\}_{j=1}^K \in \mathbb{R}^{K \times C}$, where $K$ is the number of tokens in the component's set and $C$ is the dimension of each token. To distinguish across parts, we add a learnable part identity embedding $\mathbf{e}_i \in \mathbb{R}^C$ to the tokens of each part $\mathbf{p}_i$. These embeddings are randomly initialized and optimized during training. To support the inherent permutation invariance of the parts, we shuffle the order of the parts in training. Since $\boldsymbol{z}_i$ represents the features of the $i$-th part, we construct the global 3D asset tokens simply by concatenating the tokens of all parts, i.e., $\boldsymbol{\mathcal{Z}} = \{\boldsymbol{z}_i\}_{i=1}^N \in \mathbb{R}^{NK \times C}$.

**Local-Global Denoising Transformer** We fuse information across sets of latent tokens to enable both local-level and global-level reasoning. We apply **local attention** independently to the tokens $\boldsymbol{z}_i$ of each part. This captures localized features within each part, ensuring that their internal structure remains distinct. After capturing part-level features, we apply **global attention** over the entire set of

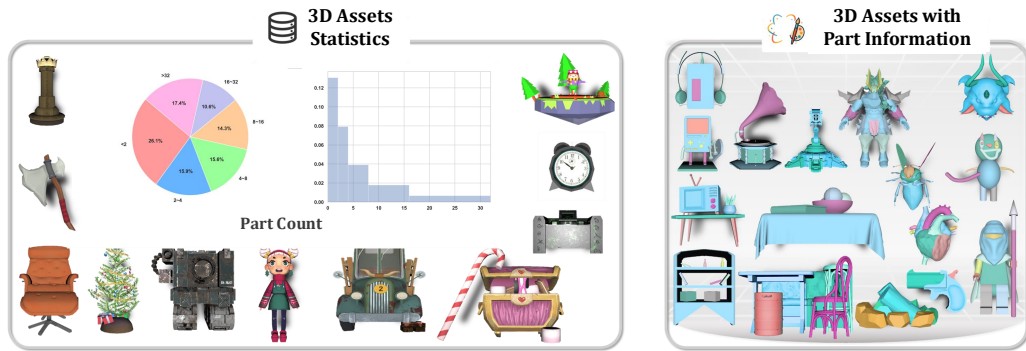

Figure 3: **Dataset Overview**. Large-scale 3D object datasets [9, 10, 11] often contain rich part annotations. The pie chart and bin chart visualize the distribution of an object's part count.

tokens $\mathcal{Z}$ to model global interactions across parts. The attention operations are defined as:

$$\mathbf{A}_i^{\text{local}} = \text{softmax}\left(\frac{\boldsymbol{z}_i \boldsymbol{z}_i^T}{\sqrt{C}}\right) \in \mathbb{R}^{K \times K}, \quad \mathbf{A}^{\text{global}} = \text{softmax}\left(\frac{\mathcal{Z} \mathcal{Z}^T}{\sqrt{C}}\right) \in \mathbb{R}^{NK \times NK}, \quad (1)$$

where $\mathbf{A}$ denotes the resulting attention maps. This hierarchical attention structure captures both fine-grained part details and global 3D context, enabling effective information exchange between local and holistic representations. We adopt the DiT architecture with long skip connections [35, 72, 73], as in TripoSG [1], and replace its original attention module with our part-global attention mechanism.

We inject DINOv2 [70] features of the condition image $\mathbf{c}$ into both levels of attention. Specifically, we use cross-attention within both the local and global attention. This dual-conditioning design enables the model to align the overall part-based composition with the input image while ensuring that each part remains semantically meaningful. Our design choices are validated in Section 4.3.

**Training Objective**    We train PARTCRAFTER by rectified flow matching [74, 75, 76, 77], which maps the noisy Gaussian distribution to the data distribution in a linear trajectory. Specifically, given an object latent $\mathcal{Z}_0 = \{z_i\}_{i=1}^N$ and a condition image $\mathbf{c}$, we perturb the latents by adding Gaussian noise $\boldsymbol{\epsilon} \sim \mathcal{N}(0, \mathbf{I})$ at a noise level $t$, yielding a noisy latent representation $\mathcal{Z}_t = t\mathcal{Z}_0 + (1 - t)\boldsymbol{\epsilon}$. The model is then trained to predict the velocity term $\boldsymbol{\epsilon} - \mathcal{Z}_0$ given the noisy latent $\mathcal{Z}_t$ at the noise level $t$, conditioned on the condition image $\mathbf{c}$, by minimizing the following objective:

$$\mathcal{L}_{\text{flow}} = \mathbb{E}_{\mathcal{Z}, \boldsymbol{\epsilon}, t}\left[\|(\boldsymbol{\epsilon} - \mathcal{Z}_0) - \mathbf{v}_\theta(\mathcal{Z}_t, t, \mathbf{c})\|^2\right], \quad (2)$$

where $\mathbf{v}_\theta$ is the velocity prediction. Importantly, the noise level $t$ is shared across all parts of the 3D scene or object to ensure consistent trajectory sampling.

## 3.3    Dataset Curation

3D object datasets such as Objaverse [9] and Objaverse-XL [71] have made millions of 3D assets available to the research community. While previous works [1, 3, 38] have primarily focused on directly utilizing these datasets [9, 10, 11, 52, 71, 78] for 3D whole-object generation, we observe that the rich part-level metadata included in many of these 3D models offers an opportunity to enable more structured 3D generation. As shown in Figure 3, over half of the objects in a subset [79] of Objaverse [9] contain explicit part annotations. These part labels often originate from artists' workflows, where objects are intentionally decomposed into semantically meaningful components to facilitate modular design, such as modeling a pair of scissors using two distinct blades. To train our model, we curate a dataset by combining multiple sources [9, 10, 11, 79], yielding 130,000 3D objects, of which 100,000 contain multiple parts. We further refine this dataset by filtering based on texture quality, part count, and average part-level Intersection over Union (IoU) to ensure high-quality supervision. The resulting dataset comprises approximately 50,000 part-labeled objects and 300,000 individual parts. For 3D scenes, we utilize the existing 3D object-composed scene dataset 3D-Front [12]. Additional dataset statistics and filtering criteria are detailed in Appendix A.

### 3.4 Implementation Details

We modify the 21 DiT blocks in TripoSG [1] by alternating their original attention processors with our proposed local-global attention mechanism. Specifically, global-level attention is applied to DiT blocks with even indices, while local-level attention is used in the odd-indexed blocks, following the long-cut strategy. We validate this architectural design in Section 4.3. PARTCRAFTER is trained on 8 H20 GPUs with a batch size of 256 by fully finetuning the pretrained TripoSG [1]. We first train a base model for up to 8 parts on our curated part-level object dataset at a learning rate of 1e-4 for 5K iterations. For part-decomposable objects, we then finetune the base model to support up to 16 parts. For object-composed scenes, we further adapt the base model to the 3D-Front [12] dataset for up to 8 objects. Both finetuning processes last for 5K iterations at a reduced learning rate of 5e-5. We include 30% monolithic objects in training for regularization. This curriculum training strategy avoids loss spikes and catastrophic forgetting during the training process. The whole training process takes about 2 days. We use 512 tokens for each part, which we find is sufficient to represent part geometry and semantics. We evaluate PARTCRAFTER on a test set of about 2K data samples.

## 4 Experiments

Our experiments aim to answer the following questions: **(1)** How does PARTCRAFTER perform in part-level reconstruction of objects and scenes compared to existing state-of-the-art models that first segment and then reconstruct parts at the object and scene level? **(2)** Can PARTCRAFTER reconstruct parts that are not visible in the image prompt? **(3)** How do results vary with different numbers of parts? **(4)** What are the contributions of design choices in our local-global denoising transformer?

**Baselines** To the best of our knowledge, PARTCRAFTER is the first work to generate 3D part-level object meshes from a single image. Recent works Part123 [4] and PartGen [5] reconstruct 3D neural fields [21, 54] from images, which are not directly comparable to our work that focuses on meshes. We consider the following baselines: **(1)** HoloPart [6] on object level, which is a concurrent work that first segments a given 3D object mesh and then completes the coarse-segmented parts into fine-grained meshes. We adapt HoloPart to our setting by utilizing TripoSG [1] to generate a mesh from an image, and apply HoloPart to get the part meshes. For fair comparison, we align the number of tokens in TripoSG with our method, that is, $N \times 512$ tokens for $N$ parts. **(2)** MIDI [7] on scene level, which reconstructs multi-instance 3D scenes using object segmentation prompts. We provide **ground truth segmentation masks** for MIDI, while PARTCRAFTER does not need any segmentation.

**Evaluation Metrics** We measure the generation result of structured 3D assets in both the global (object or scene) and part level. **(1)** Fidelity of the generated mesh. Since we do not have the correspondence between the generated and ground truth parts, we simply concatenate the parts to form a single mesh. We evaluate the fidelity of generated 3D meshes by L2 Chamfer Distance (CD) and F-Score with a threshold of 0.1. Lower Chamfer Distance and higher F-Score indicate higher similarity between the generated and ground truth meshes. **(2)** Geometry Independence of Generated Part Meshes. We use the Average Intersection over Union (IoU) to evaluate the geometry independence of generated part meshes. We compute the average IoU between each generated part by voxelizing the canonical space into $64 \times 64 \times 64$ grids. Lower IoU indicates less overlap between generated parts, thus demonstrating better part independence. The optimal metrics are reached when generated parts are non-intersecting and can be composed into an object similar to the ground truth. We report the average generation time of objects or scenes with 4 parts in the test set on an H20 GPU.

### 4.1 3D Part-level Object Generation

We evaluate the performance of PARTCRAFTER on the 3D part-level object generation task. As shown in Table 1, PARTCRAFTER outperforms HoloPart by a large margin in both object-level and part-level metrics. Given an image, PARTCRAFTER is able to generate a 3D part-decomposable mesh with high fidelity and geometry independence in seconds. HoloPart requires substantially more time to segment the object mesh, and its segmentation process suffers due to the lower geometry quality of the generated mesh compared to real artistic meshes, which hinders its performance. Notably, PARTCRAFTER surpasses our backbone model TripoSG [1] in object-level metrics, even when we align the number of tokens in TripoSG with our method. This suggests that our local-global attention mechanism enables better object-level representation learning, thereby **enhancing**

Table 1: **Evaluation on 3D Part-level Object Generation**. We report evaluation result on Objaverse [9], ShapeNet [10], and ABO [11]. We denote TripoSG* [1] as the number-of-token-aligned backbone model. Higher F-Score and lower CD, IoU indicate better results. Best results are bolded.

| 3D Part-level Generation | Objaverse [9] | | | ShapeNet [10] | | | ABO [11] | | | ↓ Time |
|---|---|---|---|---|---|---|---|---|---|---|
| | ↓ CD | ↑ F-Score | ↓ IoU | ↓ CD | ↑ F-Score | ↓ IoU | ↓ CD | ↑ F-Score | ↓ IoU | |
| Dataset | / | / | 0.0796 | / | / | 0.1827 | / | / | 0.0137 | / |
| TripoSG [1] | 0.3104 | 0.5940 | / | 0.3751 | 0.5050 | / | 0.2017 | 0.7096 | / | / |
| TripoSG* [1] | 0.1821 | 0.7115 | / | 0.3301 | 0.5589 | / | 0.1503 | 0.7723 | / | |
| HoloPart [6] | 0.1916 | 0.6916 | 0.0443 | 0.3511 | 0.5498 | 0.1107 | 0.1338 | 0.8093 | 0.0449 | 18min |
| Ours | **0.1726** | **0.7472** | **0.0359** | **0.3205** | **0.5668** | **0.0293** | **0.1047** | **0.8617** | **0.0243** | **34s** |

Table 2: **Evaluation on 3D Object-Composed Scene Generation**. We report evaluation results on 3D-Front [12], as well as on a challenging subset of 3D-Front [12] characterized by severe occlusions.

| 3D Scene Generation | 3D-Front [12] | | | 3D-Front (Occluded) [12] | | | ↓ Run Time |
|---|---|---|---|---|---|---|---|
| | ↓ CD | ↑ F-Score | ↓ IoU | ↓ CD | ↑ F-Score | ↓ IoU | |
| MIDI [7] | 0.1602 | 0.7931 | **0.0013** | 0.2591 | 0.6618 | **0.0020** | 80s |
| Ours | **0.1491** | **0.8148** | 0.0034 | **0.1508** | **0.7800** | 0.0035 | **34s** |

**the generation process through improved structural understanding**. We also observe that PARTCRAFTER performs worse on ShapeNet [10] than on Objaverse [9] and ABO [11], primarily due to the backbone's degraded performance on ShapeNet. We present qualitative results in Figure 4 to show that PARTCRAFTER **can infer parts invisible in the conditioning image** and generate 3D part-level objects. We further provide **3D object texture generation** results in Appendix D.

## 4.2 3D Object-Composed Scene Generation

We conduct experiments on 3D scene generation using the 3D-Front [12] dataset. As shown in Table 2, PARTCRAFTER outperforms MIDI [7] in reconstruction fidelity metrics. While MIDI [7] uses **ground truth segmentation masks** for evaluation, PARTCRAFTER does not require any segmentation. To further validate our method, we select a subset of 3D scenes in 3D-Front [12] with severe occlusion, where the ground truth segmentation masks cannot segment all objects. We observe that the performance of MIDI [7] degrades significantly in these cases, while PARTCRAFTER still maintains a high level of generation quality. MIDI [7] slightly surpasses PARTCRAFTER in IoU metrics, primarily due to the fact that ground truth 2D segmentation masks are used in their pipeline. Qualitative results are provided in Figure 5 to demonstrate that PARTCRAFTER can **recover complex 3D structures** and generate high-quality meshes for 3D scenes, all from a single image.

For additional qualitative results of our method, please check Appendix E and the supplementary file.

## 4.3 Ablations

We present quantitative results in Table 3 to validate the effectiveness of each component in our method. All experimental settings are consistent with our base model, using a maximum of 8 parts and 512 tokens per part, trained for 5K iterations with a learning rate of 1e-4 and a batch size of 256.

**Necessity of Local-Global Attention**    To assess the necessity of local-global attention, we conduct ablation experiments by removing either the local attention or the global attention. We observe that the model completely collapses without local attention and fails to generate any meaningful 3D mesh (Exp. 2). In contrast, removing global attention still allows the model to produce 3D meshes, but the outputs lack part decomposition, exhibit significant geometry overlap, and result in notably higher IoU values (Exp. 3). These findings demonstrate that local-global attention is essential for learning meaningful 3D representations and capturing the 3D structural relationships globally.

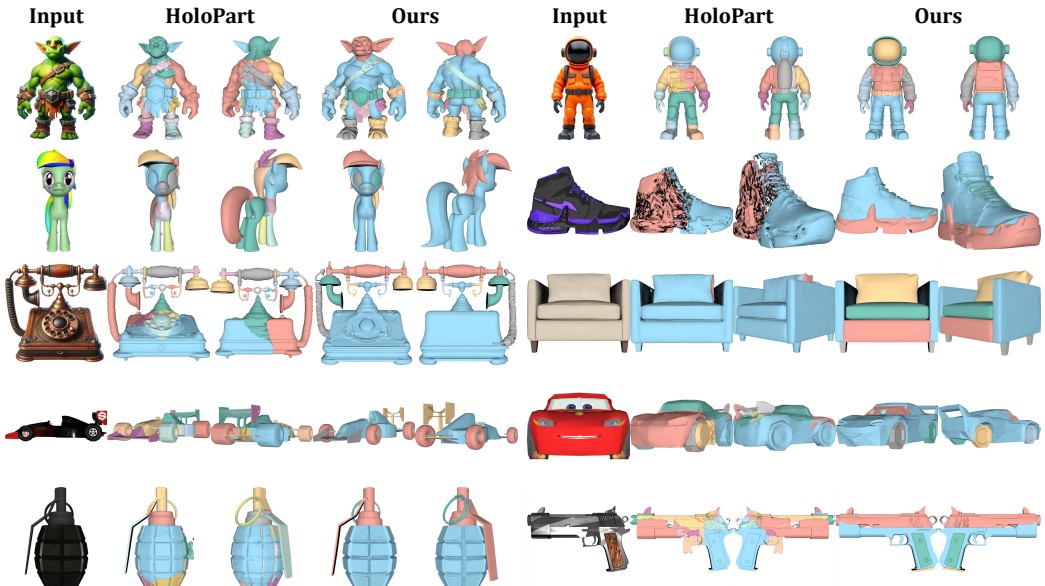

Figure 4: **Qualitative Results on 3D Part-Level Object Generation.** We present visualization results of HoloPart [6] and our method. Different colors indicate different parts of generated objects.

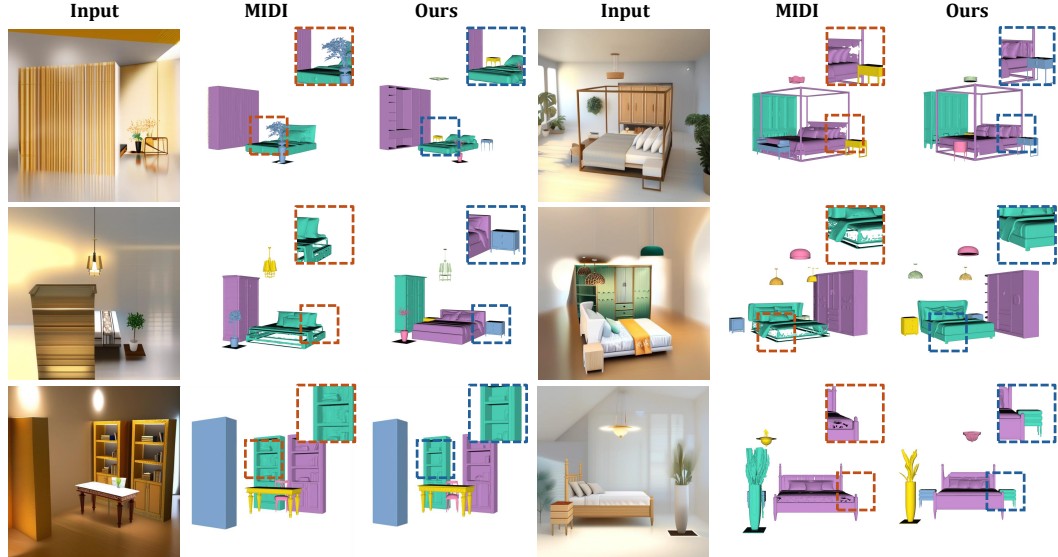

Figure 5: **Qualitative Results on 3D Scene Generation.** We present visualization results of MIDI [7] and our method. Different colors indicate different objects in the generated 3D scene.

**Necessity of Part Identity Control** We remove part identity embeddings from attentions, and the model collapses due to ambiguity in distinguishing between different parts (Exp. 1). We further explore how to incorporate control signals into the local-global attention (Exp. 4). Enabling cross-attention in the local attention improves mesh fidelity but reduces geometry independence, while enabling it in the global attention enhances geometry independence at the cost of fidelity. Based on these observations, we adopt cross-attention in both local and global attention modules to strike a balance between fidelity and independence, achieving the best overall performance.

**Order of Local-Global Attention** Since our DiT follows a U-Net-like architecture with long skip connections, we investigate how the ordering of local-global attention affects performance (Exp. 5). Across the 21 DiT blocks, we approximately balance the number of local and global attention modules. We explore three different configurations for placing global-level attention: **(1)** in the middle, **(2)** at the beginning and end, and **(3)** in an alternating pattern. Among these, the alternating configuration

Table 3: **Ablation Study for PARTCRAFTER**. We report evaluation results on Objaverse [9] dataset.

| Exp-ID | Part Emb | Self Local-Attn | Self Global-Attn | Cross Local-Attn | Cross Global-Attn | Global-Attn Order | ↓ CD | ↑ F-Score | ↓ IoU |
|---|---|---|---|---|---|---|---|---|---|
| #1 | ✗ | ✓ | ✓ | ✓ | ✓ | Middle | 0.3143 | 0.4978 | 0.1401 |
|    | ✓ | ✓ | ✓ | ✓ | ✓ | Middle | 0.1711 | 0.7374 | 0.0814 |
| #2 | ✓ | ✗ | ✓ | ✗ | ✓ | Middle | 0.4632 | 0.2327 | 0.0541 |
|    | ✓ | ✓ | ✓ | ✓ | ✓ | Middle | 0.1711 | 0.7374 | 0.0814 |
| #3 | ✓ | ✓ | ✗ | ✓ | ✗ | Middle | 0.2606 | 0.5978 | 0.1602 |
|    | ✓ | ✓ | ✓ | ✓ | ✓ | Middle | 0.1711 | 0.7374 | 0.0814 |
| #4 | ✓ | ✓ | ✓ | ✓ | ✗ | Middle | 0.1744 | 0.7334 | 0.0869 |
|    | ✓ | ✓ | ✓ | ✗ | ✓ | Middle | 0.1847 | 0.7188 | 0.0824 |
|    | ✓ | ✓ | ✓ | ✓ | ✓ | Middle | 0.1711 | 0.7374 | 0.0814 |
| #5 | ✓ | ✓ | ✓ | ✓ | ✓ | Middle | 0.1711 | 0.7374 | 0.0814 |
|    | ✓ | ✓ | ✓ | ✓ | ✓ | Sides | 0.1901 | 0.7114 | 0.0336 |
|    | ✓ | ✓ | ✓ | ✓ | ✓ | Alternating | 0.1781 | 0.7212 | 0.0518 |
| Ours | ✓ | ✓ | ✓ | ✓ | ✓ | Alternating | 0.1781 | 0.7212 | 0.0518 |

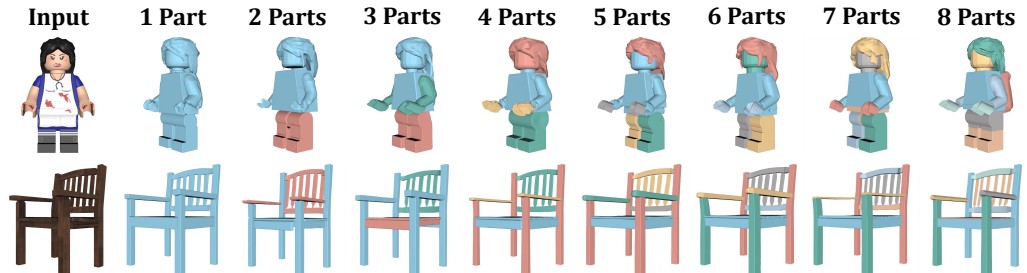

Figure 6: **Qualitative Results on Different Number of Parts.** We show that PARTCRAFTER can generate reasonable results with different granularities of part decomposition given the same image.

yields the best trade-off between local-level and global-level metrics, suggesting that this arrangement enables more effective information exchange and integration across both levels.

**Specified Number of Parts** We present qualitative results in Figure 6 to show that PARTCRAFTER can generate reasonable results with a varied number of parts given the same image. It validates that PARTCRAFTER captures 3D features hierarchically and handles different granularities of part decomposition, which can be a useful feature for downstream applications and commercial use.

## 5 Conclusion

In this work, we propose PARTCRAFTER, a novel 3D native structured generative model. Trained on our curated part-level 3D mesh dataset, PARTCRAFTER reconstructs part-level objects and scenes without relying on any 2D or 3D segmentation information that existing models need. PARTCRAFTER validates the feasibility of integrating 3D structural understanding into the generative process.

**Limitations and Future Works** PARTCRAFTER is trained on 50K part-level data, which is relatively small compared to that used to train 3D object generation models (typically millions). Future works can consider scaling up DiT training with more data of higher quality.

**Broader Impact** We conducted a foundational study in 3D computer vision. While our large-scale training may have environmental implications, we believe the benefits of advancing 3D vision justify further exploration. The importance of the problem studied is discussed in the Introduction.

## Acknowledgment

This research is supported by a grant from Bytedance PICO (No. CT20240607105793).

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

## A  Dataset Details

We collect our part-level dataset from Objaverse [9] (ODC-By v1.0 License), ShapeNet-Core[10] (Custom License), and Amazon Berkeley Objects [11] (CC-BY 4.0 License). We use a high-quality subset of Objaverse provided by LGM [79]. We filtered out objects without textures and selected those with fewer than 16 parts and a maximum IoU of less than 0.1. The resulting dataset comprises approximately 50,000 part-labeled objects and 300,000 individual parts. We adopt an additional 30,000 monolithic objects as regularization. We use the 3D-Front (CC-BY-NC 4.0 License) dataset processed by MIDI [7]. We will release our dataset under a CC-BY 4.0 license.

## B  Implementation Details

Our model builds on TripoSG [1] (MIT License). As for part-level object generation, we adapt HoloPart [6] (MIT License) into a generative pipeline. Specifically, we first generate a 3D mesh from the input image using TripoSG and then use HoloPart to generate a part-level object. We use the same 3D segmentation model, SAMPart3D [80] (MIT License), as Holopart. As for 3D scene generation, we adopt MIDI [7] (MIT License) as a baseline. We will release our code under an MIT license.

## C  Real-world Results

We train PARTCRAFTER on synthetic rendered images, which allows us to have a large and diverse dataset. To test the boundary of our model's generalization ability, we test PARTCRAFTER on real-world images from the CO3D [81] dataset. As expected, the performance was lower compared to synthetic images due to the domain gap. To address this, we explored transferring the style of real-world images to make them look more like images rendered from a graphics engine using recent image editing models, such as GPT-4o. Specifically, we use the prompt: *"Preserve all details and perform image-to-image style transfer to convert the image into the style of a 3D rendering (Objaverse-style rendering)."* We find that our model performs significantly better in the style-transferred images. Therefore, we propose a two-stage pipeline for real-world image inference: first, use an image editing model to transfer the style of the input image, and then use PARTCRAFTER to generate the 3D part-level object. As shown in Figure 7, this simple pipeline works surprisingly well.

## D  Texture Generation

As shown in Figure 8, we further generate textures for the generated 3D part-level objects using an off-the-shelf texture generation model Hunyuan3D-2 [38] (Hunyuan3D-2 License). Although Hunyuan3D-2 is trained on monolithic objects, it performs well on part-level objects. Since we know which part each vertex belongs to, we can manipulate the generated UV map accordingly, assigning textures to their corresponding parts. Therefore, our method is capable of **generating 3D objects composed of multiple distinct parts with respective textures, all from a single image**. Compared to previous methods that generate shape and texture for monolithic objects, our method produces part-aware 3D objects with distinct textures for each component. Because prior approaches treat the object as a single whole, textures often suffer from artifacts such as color bleeding between parts, and the resulting shapes may lack physically plausible part-level structure. In contrast, our method ensures more accurate texture assignment and generates 3D structures that better reflect the compositional nature of real-world objects. Thanks to these features, we believe PARTCRAFTER holds great potential for downstream applications such as real-to-simulation transfer and robotics training, where accurate and part-aware 3D representations with textures are crucial.

## E  More Results

We provide more generation results on 3D part-level object in Figure 9, 10, 11, and 12 and 3D object-composed scene in Figure 13. For better visualization results, please see the supplementary files and our project website: https://wgsxm.github.io/projects/partcrafter/.

| Input | RMBG | RMBG-Result | Stylized | Stylized-Result |
|-------|------|-------------|----------|-----------------|

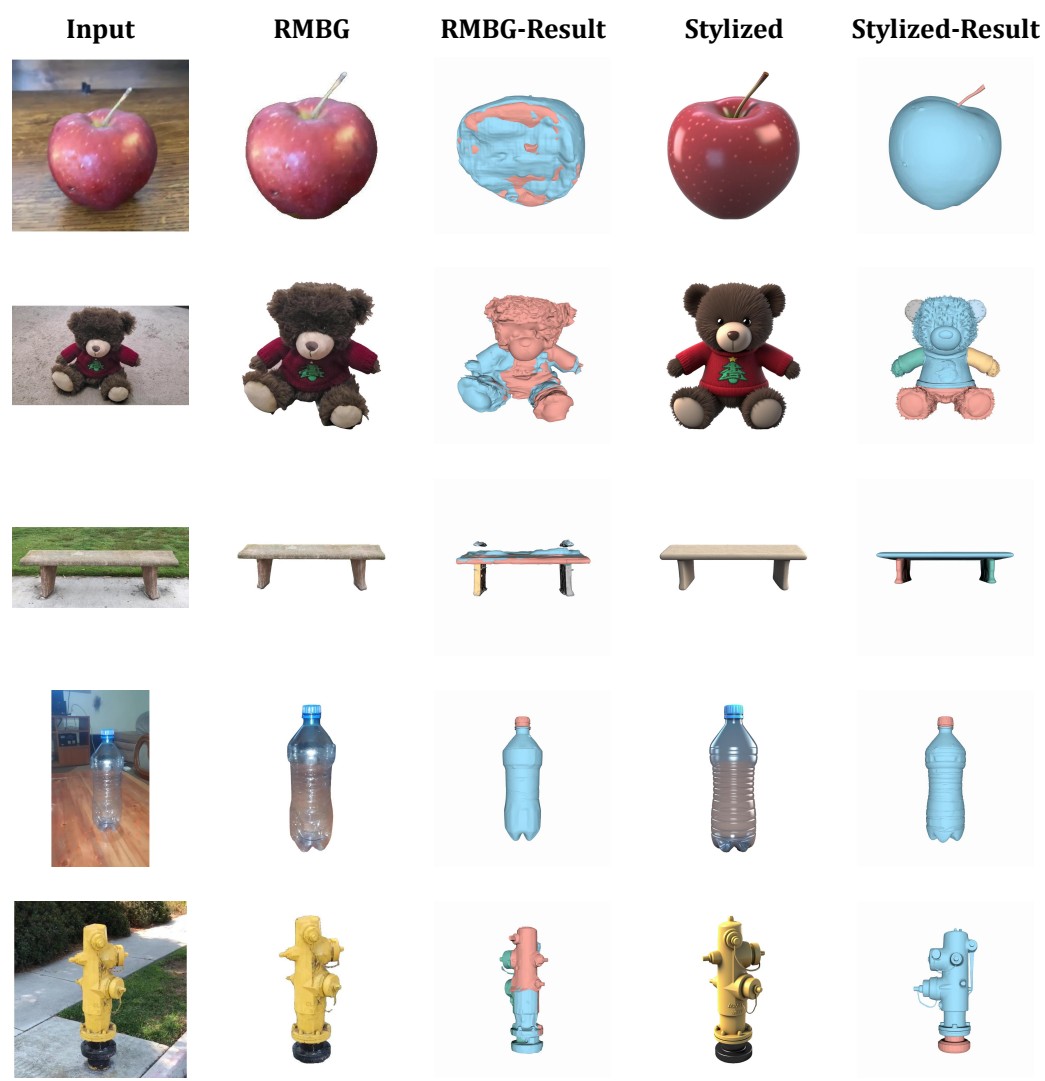

Figure 7: **Qualitative Results on 3D Part-Level Object Generation from Real-World Images.** We use GPT-4o to transfer the style of real-world images to make them look like images rendered from a graphics engine. Then, we use PARTCRAFTER to generate 3D part-level objects.

| Input | Ours | Textured | Input | Ours | Textured |
|-------|------|----------|-------|------|----------|

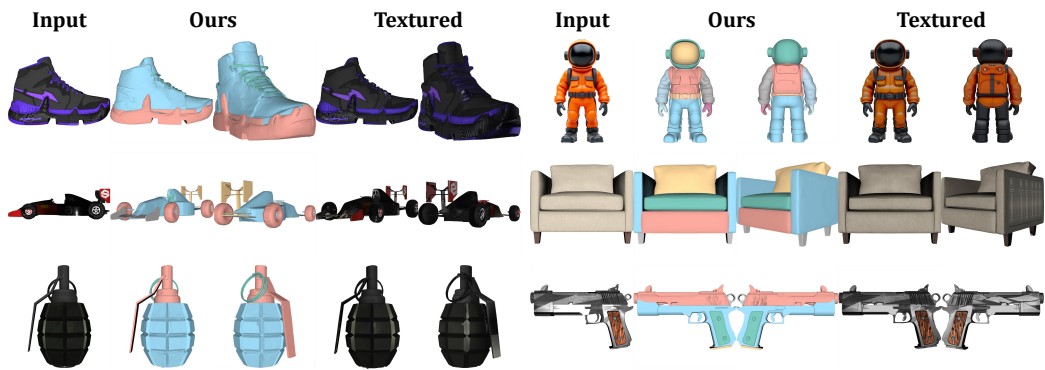

Figure 8: **Qualitative Results on 3D Textured Part-Level Object Generation.**

**Input**                    **Ours**

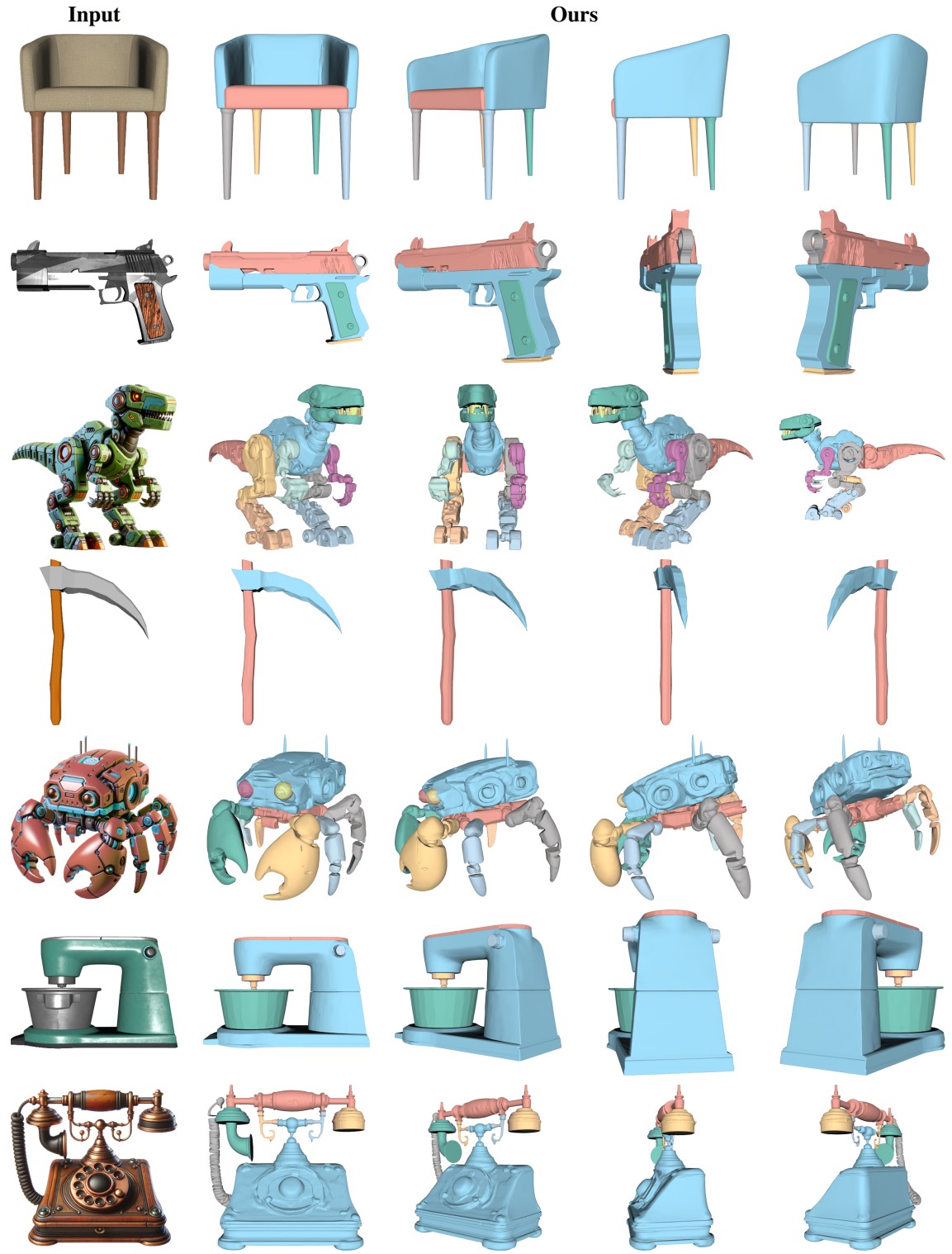

Figure 9: More results of image-conditioned 3D part-level object generation of PARTCRAFTER.

**Input**                    **Ours**

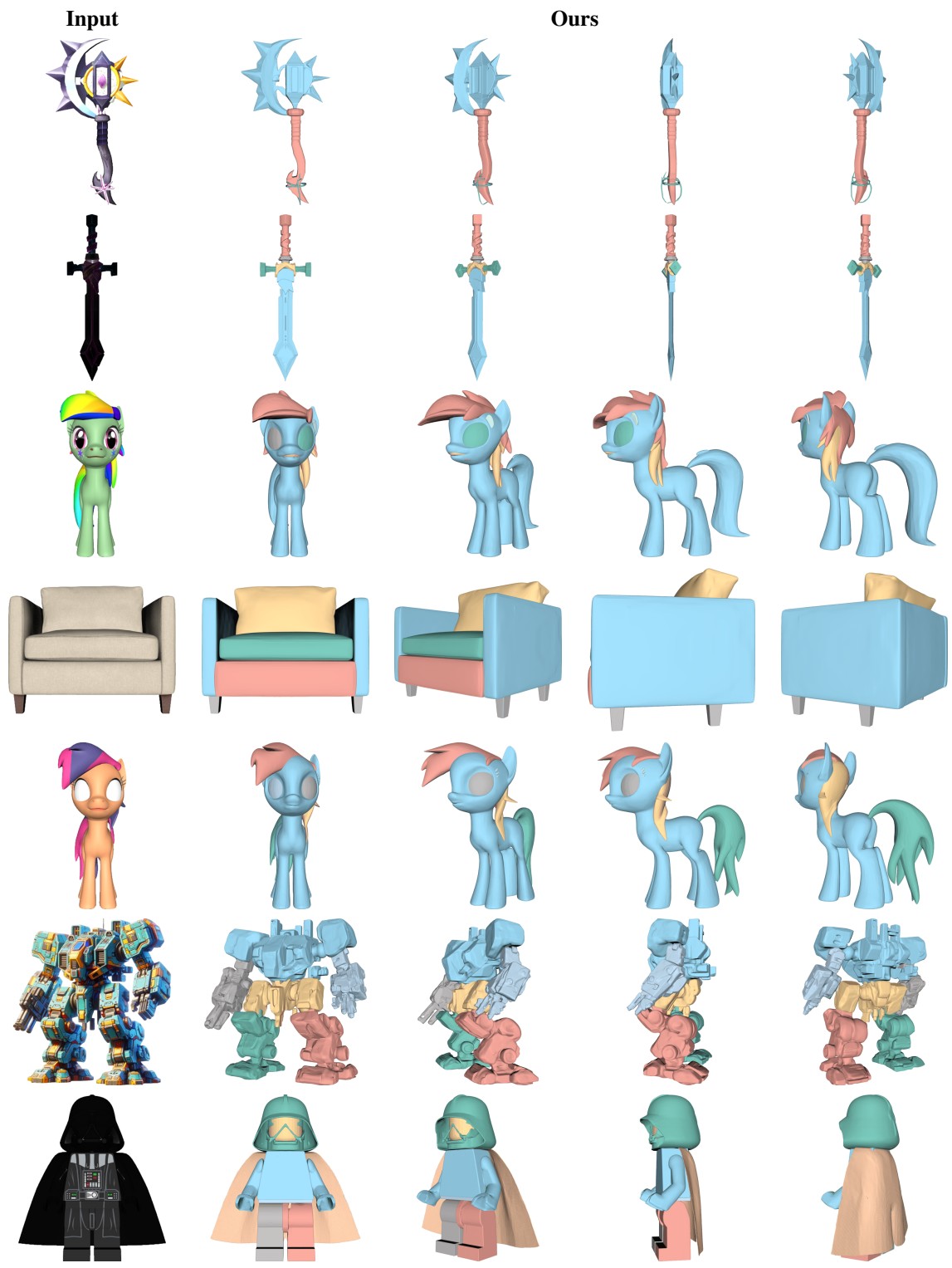

Figure 10: More results of image-conditioned 3D part-level object generation of PARTCRAFTER.

**Input**                                    **Ours**

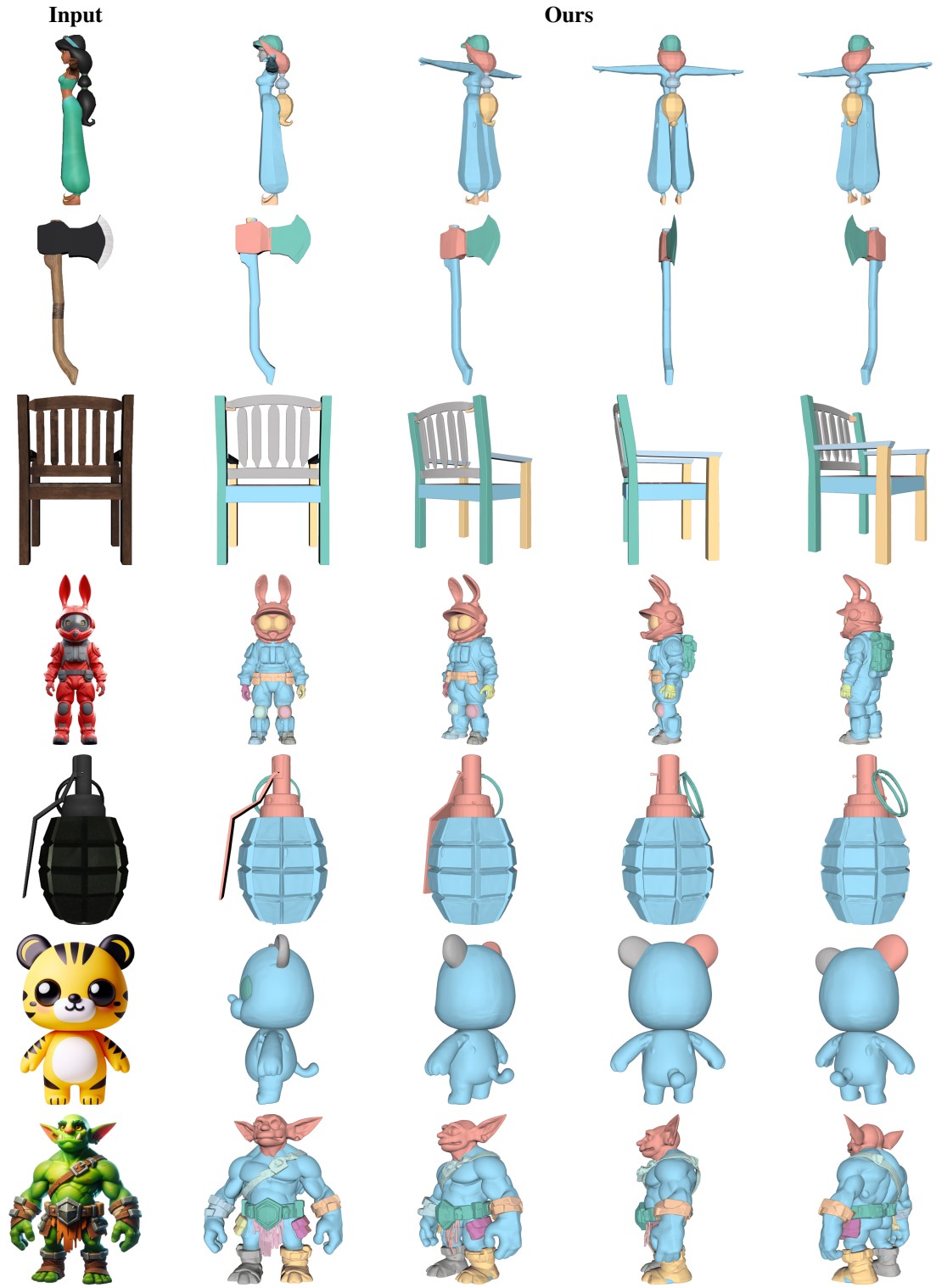

Figure 11: More results of image-conditioned 3D part-level object generation of PARTCRAFTER.

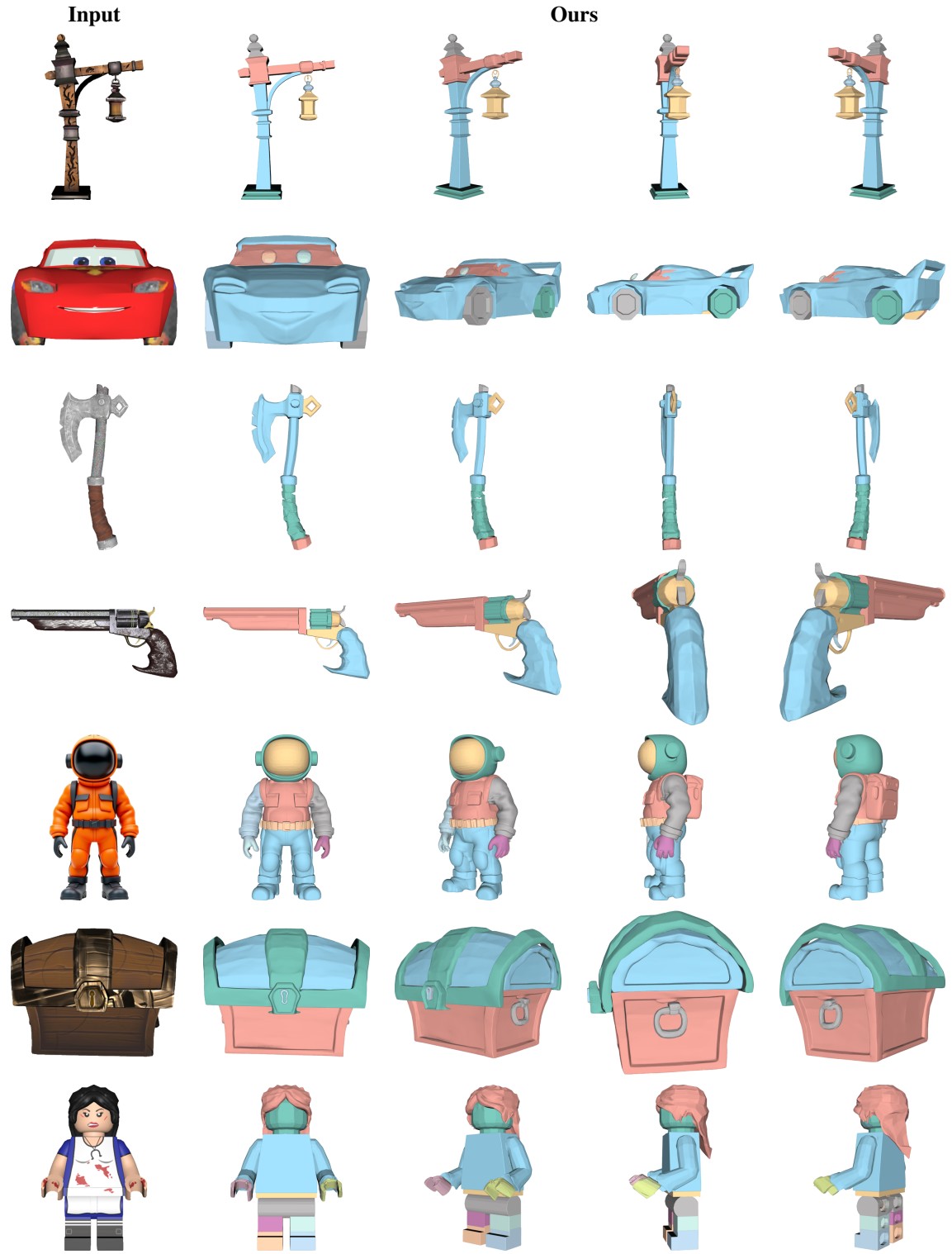

Figure 12: More results of image-conditioned 3D part-level object generation of PARTCRAFTER.

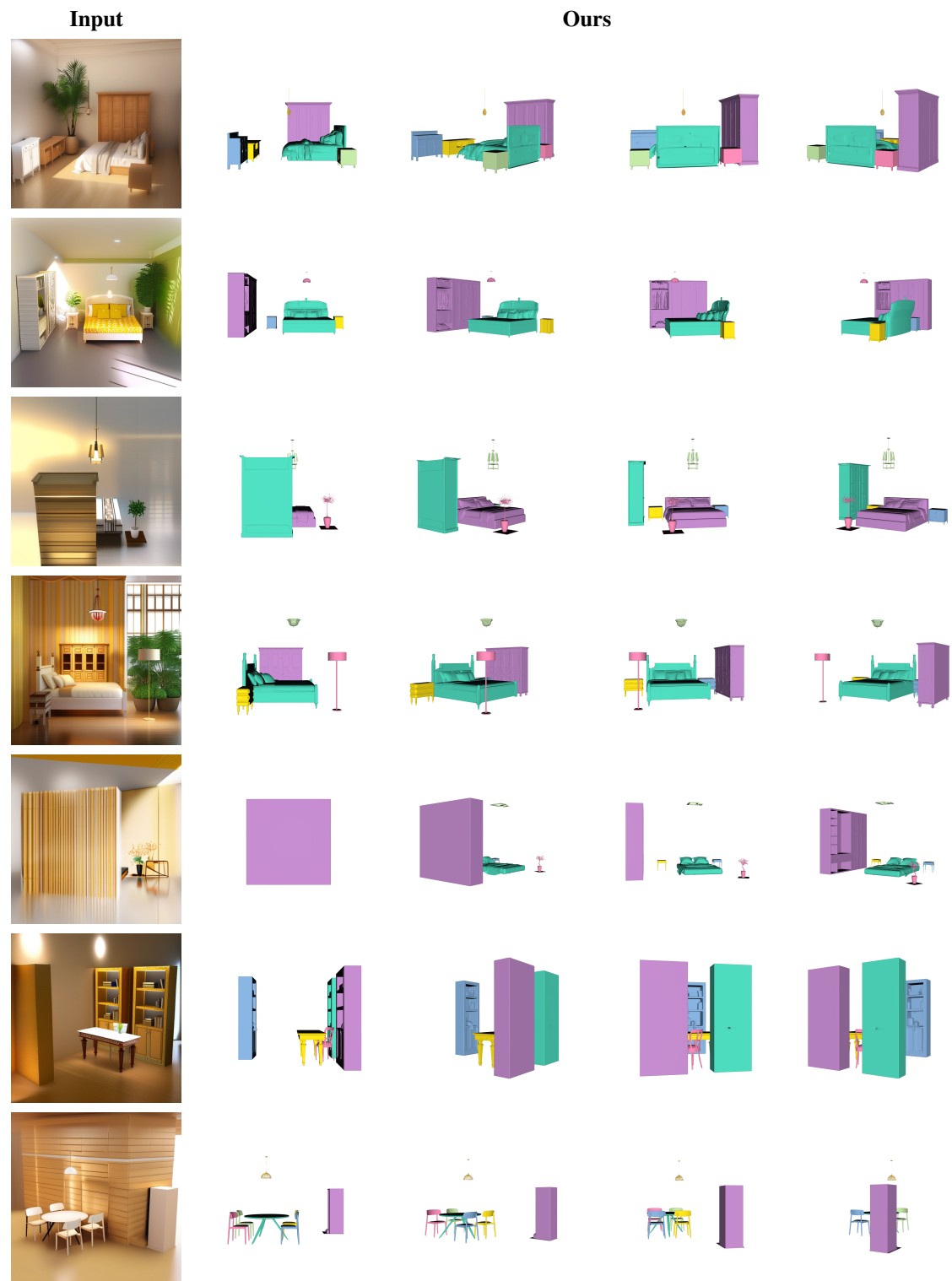

Figure 13: More results of image-conditioned 3D scene generation of PARTCRAFTER.

