# OpenReview forum: "PartCrafter: Structured 3D Mesh Generation via Compositional Latent Diffusion Transformers"
_NeurIPS.cc/2025/Conference — NeurIPS 2025 poster_

### Official Review · Reviewer_EYFc · 2025-06-29

**Clarity:** 3
**Significance:** 2
**Originality:** 2
**Rating:** 3
**Confidence:** 4

**Summary:**

The paper introduces PartCrafter, a novel structured 3D generative model that creates semantically meaningful and geometrically distinct 3D parts from a single RGB image—without requiring pre-segmented inputs. This paper proposes a unified, end-to-end architecture based on a compositional latent space and hierarchical attention. It builds on a pretrained 3D diffusion transformer, enabling generation of coherent multi-part objects and scenes. A new dataset with part-level annotations is also curated to support training. Experiments show PartCrafter outperforms previous approaches in both single object and multi-object scenarios.

**Questions:**

I would recommend the authors to consider the following questions:

1. During evaluation, how is the number of generated parts determined? Is this parameter fixed across all experiments when comparing with baseline methods, or does it vary depending on each individual object?

2. The paper currently uses IoU metrics to measure the geometric independence of generated part meshes. However, HoloPart employs IoU to evaluate the semantic accuracy of generated parts by comparing them against ground truth parts. Could the authors consider incorporating similar semantic accuracy evaluation metrics to provide a more comprehensive assessment of part generation quality?

3. Please provide additional discussion regarding the controllability of the model, particularly addressing how users can effectively control the generation process during inference.

4. It would be valuable to include more extensive results demonstrating in-the-wild generation capabilities and part editing functionality to better assess the method's practical applicability.

**Ethical Concerns:**

["NO or VERY MINOR ethics concerns only"]

**Final Justification:**

Thanks the authors for their rebuttal. However, I still have concerns regarding the controllability of the proposed methods. As mentioned by the authors, users need to specify the number of parts they wish to create. Given that there is no explicit semantic guarantee with the proposed methods, users might need multiple attempts to achieve their desired results. Additionally, non-isolated parts cannot be directly edited. Therefore, I will maintain my original rating of borderline reject.

**Limitations:**

Yes.

**Paper Formatting Concerns:**

None.

**Quality:**

2

**Strengths And Weaknesses:**

**Strengths:**
- The paper demonstrates that foundation 3D latent diffusion models (e.g., TripoSG) can effectively auto-encode 3D parts without requiring centered input meshes. This insight enables the introduction of a compositional latent space with multiple 3D part latents, allowing end-to-end generation of multi-part objects without pre-segmented inputs.
- The authors contribute a valuable dataset with part-level annotations that will benefit future research in this domain.
- The proposed method achieves superior fidelity in generated 3D parts compared to existing state-of-the-art approaches.

**Weaknesses:**
- While the end-to-end approach is innovative, the controllability of the model remains a concern. Although Figure 6 demonstrates the model's ability to vary the number of generated parts, it is unclear whether users must experiment with different part counts during inference to achieve desired results.
- The semantic consistency of generated parts appears inconsistent across different objects. For instance, Figure 10 shows that ears are sometimes connected to the head and sometimes disconnected, indicating potential issues with part boundary consistency.
- The paper lacks evaluation on in-the-wild images, limiting assessment of real-world applicability.
- The paper does not address part editing capabilities, particularly for scenarios where users wish to modify non-isolated parts (e.g., editing a foot that is connected to a leg). This represents a significant practical limitation.

---

> ### Author Rebuttal · Authors · 2025-07-30
>
> We sincerely appreciate the reviewer's insightful and valuable feedback. We are encouraged to know that you recognize the insight that pretrained VAE can be adapted to reconstruct parts and the contribution of the part-level dataset. We are also glad that you find the generation results to be impressive compared to the baselines. Below, we provide clarifications for the concerns raised. We greatly value your time and effort, and we welcome any follow-up questions or suggestions you may have.
>
> **1. The controllability of the model. It is unclear whether users must experiment with different part counts during inference to achieve desired results.**
>
> The number of parts is explicitly controllable by setting the number of latent sets $N$ in the model, where each latent set corresponds to a distinct part. This allows users to directly specify the desired part count during inference. While an input image may have a range of suitable part counts, this flexibility is intentional: it enables users to generate objects with varying levels of detail and complexity, catering to different downstream needs such as coarse vs. fine-grained editing. Our qualitative results (Figure 6) demonstrate that the model produces reasonable decompositions across different $N$, validating this design as a feature rather than a limitation.
>
> **2. The semantic consistency of the generated parts.**
>
> Intuitively, the part decomposition of a 3D object can be varied, and the model can generate different part decompositions for the same object. This also corresponds to the fact that humans may have different interpretations of what constitutes a part. Users can generate part-level objects with different seeds to obtain different part decompositions and select the one that best fits their needs.
>
> **3. In-the-wild generation capability.**
>
> Thank you for your suggestion! We agree that evaluating our model on real-world images offers a more comprehensive assessment of model performance. Currently, we train our model on synthetic rendered images, which allows us to have a large and diverse dataset. Since submission, we've tested our model on real-world images from the CO3D[1] dataset. As expected, the performance was lower compared to synthetic images due to the domain gap.
>
> To address this, we explored transferring the style of real-world images to make them look more like images rendered from a graphics engine using recent image editing models, such as GPT-4o. Specifically, we used the prompt: “Preserve all details and perform image-to-image style transfer to convert the image into the style of a 3D rendering (Objaverse-style rendering).” We found that our model performed significantly better in the style-transferred images.
>
> We are working on further enhancing real-world performance by augmenting our training data with these style-transferred images. By leveraging image editing models (e.g., GPT-4o) to adapt the rendered images to real-looking ones while preserving fine-grained structural details, we aim to narrow the synthetic-to-real domain gap and improve the model’s generalization.
>
> We will add these visualizations to our project page and the revision of this manuscript, since we cannot update the PDF during the rebuttal phase.
>
> **4. Part editing functionality, particularly for scenarios where users wish to modify non-isolated parts.**
>
> We appreciate your inquiry into the part editing functionality of our model. Our approach can naturally support transforming, substituting, and removing isolated parts. However, for non-isolated parts, the current model does not support direct editing. It may require first performing 2D image editing, such as editing a foot that is connected to a leg on the image, and then using the edited image to guide the part generation.
>
> **5. During evaluation, how is the number of generated parts determined?**
>
> We simply use the ground truth number of parts for each object in the dataset. The number of parts is not fixed, but varies depending on each individual object.
>
> **6. Use IoU metrics to measure the semantic accuracy of generated parts.**
> Thank you for the valuable suggestion. We've conducted additional experiments to evaluate the semantic accuracy using IoU metrics on Objaverse. Higher IoU indicates better semantic accuracy. The following table shows the results:
> | Model | IoU $\uparrow$ |
> |-------|------------------|
> | HoloPart | 0.4359 |
> | PartCrafter | **0.5770** |
>
> These results demonstrate that our method achieves better semantic accuracy than HoloPart, indicating the effectiveness of our part-level generation approach.
>
> We will add all the aforementioned discussions to the revision of this manuscript.
>
> ---
> [1] Reizenstein et al., Common Objects in 3D: Large-Scale Learning and Evaluation of Real-life 3D Category Reconstruction, ICCV 2021

---

> > ### Comment · Reviewer_EYFc · 2025-08-04
> >
> > Thanks the authors for their rebuttal. However, I still have concerns regarding the controllability of the proposed methods. As mentioned by the authors, users need to specify the number of parts they wish to create. Given that there is no explicit semantic guarantee with the proposed methods, users might need multiple attempts to achieve their desired results. Additionally, non-isolated parts cannot be directly edited.

---

> > > ### Author Response · Authors · 2025-08-08
> > >
> > > Dear Reviewer EYFc,
> > >
> > > We hope this message finds you well. We would like to kindly follow up regarding our recent response to your valuable comments on our submission. Your feedback is very important to us, and we would greatly appreciate any further insights or updates you could provide on the review process.
> > >
> > > Thank you very much for your time and consideration. We look forward to hearing from you.
> > >
> > > Best regards,
> > >
> > > The Authors

---

> ### Author Response · Authors · 2025-08-05
>
> We sincerely thank you once again for the thoughtful feedback and for continuing to engage with our work. We understand and appreciate the ongoing concerns regarding the controllability of the proposed method. Below, we provide clarifications for the concerns raised.
>
> **1. The Number of Parts**
>
> We address the issue of part count from two complementary perspectives:
>
> - Controllability: Our method enables explicit control over the number of generated parts via the compositional latent space design. The number of parts can be controlled by adjusting the number of latent sets $N$ in the model. Each latent set corresponds to a part, so adjusting $N$ allows users to specify how many parts they want the model to generate. **This improves controllability by allowing users to directly specify the desired granularity of decomposition.** In comparison, existing baselines such as HoloPart[1] and MIDI[2] do not support part count control, thus lacking this level of flexibility. Therefore, our method provides a meaningful contribution by introducing a controllable mechanism that is absent in prior works.
>
> - Automatability: To automate part count selection, a vision-language model (VLM), such as GPT-4o, can be integrated to perform zero-shot estimation of the number of parts in an image. For instance, given an image of a character, a VLM can infer a typical semantic decomposition (e.g., head, torso, limbs), allowing automatic inference of the appropriate part count. This offers a practical path toward fully automated part-aware generation.
>
> **2. Semantic Control**
>
> About the semantic labels, the current version of our model does not support explicit semantic control during inference. We have found that a two-stage pipeline could be used to achieve this, where the first stage generates a semantic label map for the image, and the second stage uses this label map in place of the image prompt to guide the 3D part mesh generation. We've tried this approach with GPT-4o’s image editing function for the image label map generation, and it works well. We will add the results of this two-stage pipeline to the revision of this manuscript since we cannot update the PDF during the rebuttal phase.
>
> **3. Non-isolated Parts Editing**
>
> We acknowledge that our current method does not yet support direct editing of non-isolated parts (e.g., a foot connected to a leg). This is because our primary design goal was to enable part-level generation with semantic and geometric structure, rather than fine-grained editing of connected regions. However, editing is still possible through an indirect approach: users can modify the input image (e.g., change a specific region in 2D), and the model will regenerate the corresponding 3D structure accordingly. We recognize this as a limitation and plan to address it in future work by introducing more flexible control mechanisms, such as text-based editing, which would allow users to specify desired changes to specific parts, even when those parts are not isolated.
>
> We greatly appreciate the reviewer’s constructive feedback and valuable suggestions. We would like to emphasize that our work makes a significant contribution as the first part-level 3D generative model that does not rely on pre-segmented inputs or external part segmentation models. Compared to existing baselines, our method demonstrates clear improvements in controllability, semantic accuracy, and generation quality. To further support the community, we will open-source all training and inference code, as well as the curated part-level dataset. While we acknowledge certain limitations, we believe the overall contribution and practical value of our work outweigh them. We would appreciate if the reviewer could reconsider their score, as we believe the paper is worthy of acceptance.
>
> ---
> [1] Yang et al., HoloPart: Generative 3D Part Amodal Segmentation, ICCV 2025
>
> [2] Huang et al., MIDI: Multi-Instance Diffusion for Single Image to 3D Scene Generation, CVPR 2025

---

### Official Review · Reviewer_bQqp · 2025-06-30

**Clarity:** 3
**Significance:** 3
**Originality:** 3
**Rating:** 5
**Confidence:** 5

**Summary:**

This paper proposed a method to generate 3D objects from single view image with part-level decomposition. There are two key innovations: (1) A compositional latent space where each 3D part is represented by a set of disentangled latent tokens; (2) A hierarchical attention mechanism that enables structured information flow both within individual parts and across all parts. The experiment shows the good performance of the proposed method.

**Questions:**

1. The paper only presented good cases. What would be the border of the model capability?
2. Please explain in more details the evaluation process for Table 1. Specifically, are there inner structures in the GT mesh, and does this incurs unfairness that make TripoSG have disadvantage in metrics.

**Ethical Concerns:**

["NO or VERY MINOR ethics concerns only"]

**Final Justification:**

The authors addressed my concerns comprehensively. I lean to accept this paper.

**Limitations:**

No obvious limitation.

**Quality:**

3

**Strengths And Weaknesses:**

Strengths:
1. The method show promising results
2. The ablation is comprehensive.

Weakness:
1. concerns about data leak:
1.1 the method is initialized with TripoSG, which utilized many data sources for training their model including objaverse and objaverse xl, so that even the author made a test split from their filtered data, the objects are highly likely have been part of the training set of its pretrained backbone model.
1.2 Weapon, axe, chairs, sword are really very abundant in the objaverse dataset, and many of such objects are just the same whitebox mesh with different textures, so that it is not safe and convincing to demonstrate the effectiveness with these objects.

2. Fairness of evaluation: The paper reported 3D Part-level Object Generation result on Objaverse (Table 1). It includes TripoSG while TripoSG was trained on Objaverse. Therefore, it is weird if the method can perform better than TripoSG*. However, the author explains in the paper that the evaluation was performed on part-level objects, which implies that the GT objects all have inner (occluded) structures. However, TripoSG was trained on watertight objects with no inner structure. This may present a significant unfairness in evaluation.

3. It would be good if a pipeline figure can be self-contained. However, it is not clean what does Part-decomposable object and Monolithic object means and what is their purpose if we only read the figure and its caption.

4. insufficient evaluation: following weakness 1, the presented visual results are mainly cartoon-style, game-style, or simple, static, convex and common real-world objects. This may be insufficient to demonstrate that the proposed method is truly effective on real use cases. Usually, many method claimed good performance fails if being tested out of the above scoop. To demonstrate that the proposed method is really effectiveness, the author should present more examples, including 1) images captured in real-world, 2) objects with some dynamics (e.g. a realistic-style image about a dog running to the camera); 3) objects with complex spatial inter-part relationship.

---

> ### Author Rebuttal · Authors · 2025-07-30
>
> We sincerely appreciate the reviewer's insightful and valuable feedback. We are encouraged to know that you recognize the innovation of our proposed compositional latent space and the local-global attention mechanism. Below, we provide clarifications for the concerns raised. We greatly value your time and effort, and we welcome any follow-up questions or suggestions you may have.
>
> **1. Data Leakage between the train and test sets.**
>
> - The method is initialized with TripoSG, so that even if the author made a test split from their filtered data, the objects are highly likely to have been part of the training set of its pretrained backbone model.
>
>    We sincerely thank you for your insightful observation. TripoSG[1] is trained on Objaverse, Objaverse-XL, and ShapeNet. In the manuscript, we have provided results on ABO (object level) and 3D-Front (scene level). We show our method works well on these datasets, which are not part of the training set of TripoSG. We will only focus on these evaluations since they are not contaminated. Meanwhile, our baselines (HoloPart and MIDI) are also fine-tuned from TripoSG, so comparing with them is still fair.
>
> - Weapons, axes, chairs, and swords are really abundant in the objaverse dataset, so it is not safe and convincing to demonstrate the effectiveness with these objects.
>
>     We agree that these objects are abundant in the Objaverse dataset. We've provided results on a wide range of objects, such as human characters, vehicles, and furniture, to demonstrate the effectiveness of our method in the Figures and Supplementary website.
>
> **2. It is weird if the method can perform better than the number-of-token-aligned TripoSG. Is the evaluation fair since TripoSG is trained on watertight objects with no inner structure?**
>
> Thank you for raising this important point. We acknowledge that the presence of internal structures in the ground-truth objects affects the evaluation due to the symmetric nature of the Chamfer Distance. This oversight was on our part. To address this, we are currently re-evaluating our method against TripoSG* at the whole-object level, after removing internal structures from both the ground-truth meshes and the generated part-based meshes. This will ensure a fair comparison at the object level. The updated results will be included in the revised manuscript.
>
> We would also like to clarify that our model is specifically designed for part-based generation. We reported the object-level scores to provide a point of reference against a widely recognized baseline. We appreciate your feedback, which has helped us identify and correct this evaluation issue.
>
> **3. In the pipeline figure, it is not clear what a Part-decomposable object and a Monolithic object mean, and what their purpose is if we only read the figure and its caption.**
>
> We apologize for the confusion. In the pipeline figure, "Part-decomposable object" refers to objects that are composed of multiple distinct parts, and "Monolithic object" refers to objects that are not decomposed into parts. The purpose of this distinction is to highlight that our method can handle part-level generation compared to many previous monolithic generation methods. We'll add "PartCrafter can generate structured 3D models with multiple semantic parts (Part-decomposable objects), in contrast to previous approaches that generate entire shapes as single entities (Monolithic objects)" to the caption of the pipeline figure.
>
> **4. Insufficient evaluation.**
> - Real-world images.
>
>     Thank you for your suggestion! We agree that evaluating our model on real-world images offers a more comprehensive assessment of model performance. Currently, we train our model on synthetic rendered images, which allows us to have a large and diverse dataset. Since submission, we've tested our model on real-world images from the CO3D[2] dataset. As expected, the performance was lower compared to synthetic images due to the domain gap.
>
>     To address this, we explored transferring the style of real-world images to make them look more like images rendered from a graphics engine using recent image editing models, such as GPT-4o. Specifically, we used the prompt: “Preserve all details and perform image-to-image style transfer to convert the image into the style of a 3D rendering (Objaverse-style rendering).” We found that our model performed significantly better in the style transferred images.
>
>     We are working on further enhancing real-world performance by augmenting our training data with these style-transferred images. By leveraging image editing models (e.g., GPT-4o) to adapt the rendered images to real-looking ones while preserving fine-grained structural details, we aim to narrow the synthetic-to-real domain gap and improve the model’s generalization.
>
>     We will add these visualizations to our project page and the revision of this manuscript, since we cannot update the PDF during the rebuttal phase.
>
> - Objects with dynamics.
>
>     Currently, our method focuses on static objects, an assumption widely adopted in many 3D generation approaches. We acknowledge that dynamic objects lie beyond the scope of the current work, and we leave their exploration to future research.
>
> - Objects with complex spatial structure.
>
>     Since spatial structural complexity is difficult to define and quantify directly, we use the number of parts as a proxy to represent different levels of complexity. To this end, we conduct experiments on Objaverse to evaluate the model's performance under varying numbers of parts. The following table shows the results:
>
>     | Number of Parts | CD $\downarrow$ | F-Score $\uparrow$ | IoU $\downarrow$ |
>     |-------------|--------|---------|---------|
>     |           2 | 0.1310 |  0.8037 |  0.0418 |
>     |           4 | 0.1476 |  0.7735 |  0.0249 |
>     |           6 | 0.1352 |  0.7924 |  0.0198 |
>     |           8 | 0.1401 |  0.7897 |  0.0227 |
>     |          10 | 0.1319 |  0.7908 |  0.0211 |
>     |          12 | 0.1366 |  0.7837 |  0.0200 |
>     |          14 | 0.1528 |  0.7756 |  0.0221 |
>     |          16 | 0.1417 |  0.7881 |  0.0198 |
>
>     From the table, we observe that the model's performance remains relatively stable across different numbers of parts, suggesting robustness to varying levels of spatial complexity.
>
>
> **5. The border of the model's capability.**
>
> We analyze the border of the model's capability in two aspects: (1) the capability of the current model by failure case analysis and (2) the potential of the method by scaling up the model.
>
> - The capability of the current model.
>
>     In our experiments, the failure cases are mainly due to the following three reasons. (1) Unsatisfactory geometry details, such as nonsmooth surfaces or floating parts. This is often caused by insufficient denoising steps or too small number of tokens. (2) Unsatisfactory part decomposition, such as generating parts that are not semantically meaningful. This is often caused by the input image not providing enough information for the model to infer the correct part decomposition, such as an input image of a rock with no clear part structure. (3) Out-of-distribution images, such as real-world images, can also lead to failure cases. Currently, we train our model on synthetic rendered images, which allows us to have a large and diverse dataset. We've conducted experiments on real-world images. The results are not as good as those on synthetic images due to the domain gap. We have solved this issue by doing style transfer through automated image editing in order to transfer the real-world images into in-domain 3D-rendering style images. We find this strategy works well.
>
>     Despite these limitations, our method demonstrates strong performance on a wide range of objects, including human characters, vehicles, and furniture. We hope PartCrafter can serve as a solid foundation for future research in part-level 3D generation. And we will release the training, inference code, the model weights, and the part-level dataset to facilitate future research.
>
> - The potential of the method.
>
>     Due to the resource constraints and the time limit, the submission version of our method is trained on a subset of Objaverse without fine-tuning the pretrained VAE. We believe that the method has great potential to be scaled up. By using more training data, fine-tuning the pretrained VAE, and scaling up the model, we can achieve even better performance.
>
>     In the submission version, we set the number of tokens to 512 and the denoising steps to 50. Since submission, we have scaled the model to 1024 tokens and 100 denoising steps, which improves the generation quality and geometry details. The following table shows the evaluation results on the Objaverse dataset with the scaled-up model:
>
>     | Model | CD $\downarrow$ | F-Score $\uparrow$ | IoU $\downarrow$ |
>     |-------|-----------------|--------------------|------------------|
>     | Before Scaling | 0.1726          | 0.7472             | 0.0359           |
>     | After Scaling  | **0.1360**          | **0.7942**             | **0.0315**           |
>
> We will add all the aforementioned discussions to the revision of this manuscript.
>
> ---
> [1] Li et al., TripoSG: High-Fidelity 3D Shape Synthesis using Large-Scale Rectified Flow Models, arXiv 2025
>
> [2] Reizenstein et al., Common Objects in 3D: Large-Scale Learning and Evaluation of Real-life 3D Category Reconstruction, ICCV 2021

---

> > ### Comment · Reviewer_bQqp · 2025-08-01
> >
> > Thank you for the response. I have read it as well as other reviews. It well addressed my concerns. I hope the authors keep their promise and add the aforementioned discussions to the revision of this manuscript.

---

> > > ### Author Response · Authors · 2025-08-04
> > >
> > > We sincerely thank the reviewer for the time and effort devoted during both the review and rebuttal phases. Your constructive comments have been extremely helpful in improving the quality of our manuscript. As promised, we will revise the manuscript accordingly in future versions and release the source code.

---

### Official Review · Reviewer_BRws · 2025-07-01

**Clarity:** 3
**Significance:** 2
**Originality:** 3
**Rating:** 4
**Confidence:** 4

**Summary:**

This paper proposes the first part-aware native 3D generator. The model builds upon a pre-trained whole-object generator and introduces two key innovations: a compositional latent space and a local-global attention mechanism. In addition, the authors curate a 3D dataset with part annotations to facilitate training. Experimental results demonstrate the model's superiority over traditional two-stage pipelines in terms of both efficiency and part segmentation quality.

**Questions:**

- How does the number of parts affect the model’s performance in terms of both generation quality and computational cost? A more thorough analysis would be helpful.

- Could you provide more details on how part annotations were mined from Objaverse? Given that its file structure is artist-defined and hierarchical, how was this handled to ensure consistency?

**Ethical Concerns:**

["NO or VERY MINOR ethics concerns only"]

**Final Justification:**

This is a novel method for part-aware 3D object generation. I believe it will serve as a strong baseline for future work and be beneficial to the community. Therefore, I am inclined to recommend acceptance.

**Limitations:**

yes

**Quality:**

2

**Strengths And Weaknesses:**

## Strengths

- The paper presents a novel approach to part-aware 3D generation. Unlike conventional two-stage pipelines, the proposed method generates part-decomposable objects in an end-to-end fashion. This design enhances both efficiency and segmentation quality.

- The experimental evaluation is comprehensive, covering both object- and scene-level domains. The results show significant improvements in part segmentation quality compared to prior methods.

- The paper is well-organized and clearly written, making it easy to follow.

## Weaknesses

- The paper lacks a detailed analysis of how the number of parts affects both time cost and generation quality. For example, in Figure 6 (Lego figurine), the hand shapes deteriorate in the 2-part and 4-part settings, indicating instability.

- The model’s ability to handle occluded or invisible parts at the object level remains uncertain. For instance, in Figure 2, the left arm in the conditioning image is partially occluded by fire, and the generated 3D asset fails to reconstruct a complete left arm.

- The paper lacks failure cases.

- The choice of evaluation metric could be improved. While IoU is used to evaluate part-level quality, it may not capture semantic correctness. For example, parts with low IoU might be semantically meaningless.

---

> ### Author Rebuttal · Authors · 2025-07-30
>
> Thank you for your insightful and valuable feedback. We are encouraged to know that you recognize the novelty of our proposed method, including the design of the one-stage end-to-end pipeline, the compositional latent space, and the local-global attention mechanism. Below, we provide clarifications for the concerns raised. We greatly value your time and effort, and we welcome any follow-up questions or suggestions you may have.
>
> **1. How the number of parts affects both time cost and generation quality?**
> - Computational cost.
>
>     We conducted experiments to analyze the impact of the number of parts on computational cost. As shown in the table below, the time cost increases with the number of parts. The inference time is still far lower than that of HoloPart (about 18 minutes), even with 16 parts, demonstrating the efficiency of our method.
>
>     | Number of Parts | Inference Time (s) $\downarrow$ |
>     |-----------------|---------------------|
>     | 1               | 11.67                |
>     | 2               | 18.91                |
>     | 4               | 32.99                |
>     | 8               | 69.43                |
>     | 16              | 131.1                |
>
> - Generation quality.
>
>     The generation quality depends on many factors, including the complexity of the input image, the number of parts, the number of tokens, the denoising steps, and the classifier-free guidance scale.
>
>     As shown in Figure 6, the quality roughly remains consistent across different numbers of parts, with some variations in detail and structure. Intuitively, an input image should have a range of suitable numbers of parts for optimal generation quality. If the number of parts is too small, the model may not be able to capture the full complexity of the object, leading to lower quality generation. Conversely, if the number of parts is too large, it may introduce unnecessary complexity and noise, also degrading quality.
>
>     We have found that the number of tokens and denoising steps has a more significant impact on generation quality than the number of parts. In the submission version, we set the number of tokens to 512 and the denoising steps to 50. Since submission, we have scaled the model to 1024 tokens and 100 denoising steps, which improves the generation quality and geometry details, such as the hands and fingers of the generated human characters. The following table shows the evaluation results on the Objaverse dataset with the scaled-up model:
>
>     | Model | CD $\downarrow$ | F-Score $\uparrow$ | IoU $\downarrow$ |
>     |-------|-----------------|--------------------|------------------|
>     | Before Scaling | 0.1726          | 0.7472             | 0.0359           |
>     | After Scaling  | **0.1360**          | **0.7942**             | **0.0315**           |
>
> - Incomplete arm mesh in Figure 2
>
>     Thank you for pointing this out. We conducted further experiments and identified two contributing factors: (1) a large classifier-free guidance (CFG) scale, which causes the model to overfit to the input image, and (2) an insufficient number of tokens and denoising steps, which limits the refinement of the generated mesh.
>     This issue does not occur in the scaled-up version of our model, as mentioned earlier.
>
> We will include a discussion of these findings in the revised manuscript, as we are unable to update the PDF during the rebuttal phase.
>
>
> **2. The model’s ability to handle occluded or invisible parts at the object level.**
>
> **The model can generate occluded, invisible or fully internal to the object parts.** In the car example of Figure 4, not all wheels are visible in the input image, yet the model successfully generates all four wheels. In the first example in the "Image to 3D Part-Level Object Generation" section of the supplementary website (a LEGO character), the model generates slot-like structures that are inside the object.  This demonstrates that the model can infer and generate parts that are not directly visible in the input image, leveraging its understanding of object structure and context.
>
> We are working on benchmarking visible versus internal versus occluded part 3D reconstructions, and we will add those in the revised manuscript per your request.
>
> **3. Failure cases.**
>
> Thank you for pointing this out! In our experiments, the failure cases are mainly due to the following three reasons. (1) Unsatisfactory geometry details, such as nonsmooth surfaces or floating parts. This is often caused by insufficient denoising steps or too small number of tokens. (2) Unsatisfactory part decomposition, such as generating parts that are not semantically meaningful. This is often caused by the input image not providing enough information for the model to infer the correct part decomposition, such as an input image of a rock with no clear part structure. (3) Out-of-distribution images, such as real-world images, can also lead to failure cases. Currently, we train our model on synthetic rendered images, which allows us to have a large and diverse dataset. We've conducted experiments on real-world images. The results are not as good as those on synthetic images due to the domain gap. We have solved this issue by doing style transfer through automated image editing in order to transfer the real-world images into in-domain 3D-rendering style images. We find this strategy works well.
>
> We will add visualizations of all failure cases to our project page and the revision of this manuscript, since we cannot update the PDF during the rebuttal phase.
>
> **4. The choice of the IoU metric for evaluating the geometric independence of generated part meshes.**
>
> Thank you for your valuable feedback. We agree that each metric has its limitations. To ensure a comprehensive evaluation, we employ multiple metrics beyond IoU. As shown in Table 1, we report performance using Chamfer Distance and F-score, in line with established practices in part-based 3D reconstruction, such as Holopart and MIDI. We would be happy to include any additional metrics you may suggest.
>
> **5. Details on how part annotations were mined from Objaverse? How was this handled to ensure consistency?**
>
> We simply use `parts = mesh.dump()` where `mesh` is a `trimesh.Scene` object. The `dump()` method in trimesh is used to decompose a `trimesh.Scene` object into multiple `trimesh.Trimesh` objects, each representing a part of the original mesh. Internally, a `Scene` consists of multiple geometries, each with its own transformation (position, rotation, scale) and potentially corresponding to a semantically meaningful component of the overall object. When `dump()` is called, it iterates through all geometries in the scene, applies their transformations to bring them into world coordinates, and returns each one as an individual mesh.
>
> Intuitively, artists construct 3D models using semantically meaningful parts (e.g., a human figure with arms, hair, and torso) for modular design, which ensures part decompositions align with intuitive, real-world divisions. We do not perform consistency checks on the part annotations, as humans may have different interpretations of what constitutes a part. Thus, we let the model learn from the existing structure in Objaverse, which results in more diverse, realistic, and human-like part decompositions.
>
> We will add all the aforementioned discussions to the revision of this manuscript.

---

### Official Review · Reviewer_ynhN · 2025-07-01

**Clarity:** 3
**Significance:** 3
**Originality:** 3
**Rating:** 5
**Confidence:** 3

**Summary:**

This paper proposes a method to unify the architecture for generating decomposable meshes conditioned on an image. Concretely, the authors propose a one-stage pipeline to generate part mesh instead of following a first segment, then repair the parts paradigm. The method is based on the diffusion network, and the proposed compositional latent space to enable global and local attention for information within and across parts. The author also curated a part-based dataset from open-source datasets, as claimed to be part of the contribution. The authors also compare both single mesh and scene-based part generation with SoTA methods like HoloPart.

**Questions:**

--The part dataset is constructed by parsing the node, how to make sure the node grouping is meaningful?

--In Figure 6, it shows the model can generate variable length of parts, is the training also randomly select the number of parts, or does it learn to adapt to the number of parts?

**Ethical Concerns:**

["NO or VERY MINOR ethics concerns only"]

**Final Justification:**

The author's response addresses some of my concerns, and this work is promising for the future part-based 3D generative model; hence, I decided to increase my original rating.

**Limitations:**

Yes

**Quality:**

3

**Strengths And Weaknesses:**

Strengths:
--The work adopts TripoSG’s 3D VAE representation, which associates input point cloud locations to a set of latents; thus, the parts can use a shared canonical space instead of renormalizing into a canonical space separately. This representation allows the model to generate parts with transformation baked in and does not need to predict additional transformation information for each part.

--The adaptation of global and local attention to model global part interaction and per-part details is novel. The author also provides a detailed ablation study on the importance of using dual attention.

Weakness:
--The author adapts the TripoSG VAE representation and found that even when the VAE is trained on a single mesh, it can still reconstruct part-based meshes. This is a strong assumption, but there is no validation in the experiment to prove it. It would be great to show the reconstruction accuracy for parts with TripoSG VAE to validate the assumption.

--The current pipeline cannot control the number of parts and the semantic labels, which are important in practice.

---

> ### Author Rebuttal · Authors · 2025-07-30
>
> We sincerely appreciate the reviewer's insightful and valuable feedback. We are encouraged that you recognize the novelty of our proposed method, including the adaptation of the pretrained VAE and the local-global attention mechanism. Below, we provide clarifications for the concerns raised. We greatly value your time and effort, and we welcome any follow-up questions or suggestions you may have.
>
> **1. Validation of the TripoSG VAE's ability to reconstruct parts.**
>
> Thank you for pointing this out! Per your suggestion, we have conducted additional experiments to validate the TripoSG VAE's ability to reconstruct parts while being trained on objects. We calculate the Chamfer Distance (CD) and F-Score between the reconstructed mesh and the ground truth mesh. Lower CD and higher F-Score indicate better reconstruction quality. We conduct evaluations on both the whole-object level and the part level. The following table shows the results:
>
> | Level   |   CD $\downarrow$ |   F-Score $\uparrow$ |
> |---------|-------------------|----------------------|
> | Object  |            0.0153 |               0.9824 |
> | Part    |            0.0413 |               0.9398 |
>
> The CD and F-Score on the part level are only slightly worse than those on the whole-object level. We will add this experiment in the main paper. It's fair since we do not fine-tune the pretrained VAE, and the VAE is trained on whole objects. The results indicate that the TripoSG VAE can effectively reconstruct parts, achieving a close performance comparable to the whole object.
>
> **2. Control of the number of parts and the semantic labels during inference.**
>
> The number of parts can be controlled by adjusting the number of latent sets $N$ in the model. Each latent set corresponds to a part, so adjusting $N$ allows users to specify how many parts they want the model to generate.
>
> About the semantic labels, the current version of our model does not support explicit semantic control during inference. We have found that a two-stage pipeline could be used to achieve this, where the first stage generates a semantic label map for the image, and the second stage uses this label map in place of the image prompt to guide the 3D part mesh generation. We've tried this approach with GPT-4o’s image editing function for the image label map generation, and it works well. We will add the results of this two-stage pipeline to the revision of this manuscript since we cannot update the PDF during the rebuttal phase.
>
> **3. How to ensure the part decomposition is meaningful in the dataset?**
>
> We build our part-level dataset by using existing part annotations in Objaverse. Intuitively, artists construct 3D models using semantically meaningful parts (e.g., a human figure with arms, hair, torso) for modular design. This ensures part decompositions align with intuitive, real-world structures and their semantic components.
>
> **4. Is the training randomly selecting the number of parts, or does it learn to adapt to the number of parts?**
>
> While the model does not explicitly predict the number of parts, it is trained with objects that contain different numbers of parts. The number of parts is not a fixed hyperparameter of our model but rather a variable that can change per datapoint. Thanks to the use of attention layers, the model can handle varying numbers of parts in the form of a varying number of latent sets in its input. In each training step, the model processes the input object with its actual number of annotated parts, allowing it to learn to handle different configurations. In other words, **we train a single model that can handle a varying number of parts per data point both during training and inference.**
>
> During inference, the number of parts can be controlled by controlling the number of latent sets $N$ in the model. Each latent set corresponds to a part, so adjusting $N$ allows users to specify how many parts they want the model to generate.
>
> We will add all the aforementioned discussions to the revision of this manuscript.

---

> > ### Comment · Reviewer_ynhN · 2025-08-08
> >
> > Thank you for answering my questions. The VAE reconstruction results for parts and objects are interesting, and it seems the part-based reconstruction performance can be further improved with fine-tuning. I appreciate the efforts the author made to control the semantic meaning of the parts with image-based labels, but it requires an external model for labels and can cause extra errors. It would be great if the authors could make the controllability explanation clearer in the main text, and it is fine to leave some of them for future work.
> >
> > With the above, it addressed some of my concerns, and I agree to raise my score.

---

> > > ### Author Response · Authors · 2025-08-08
> > >
> > > We sincerely thank you for the constructive suggestions, which have helped us further refine and improve the clarity of our paper. We agree that finetuning the VAE could yield better part-level reconstruction results. We also acknowledge the limitations of the two-stage semantic control and appreciate your understanding. We are glad to hear that you are inclined to accept our work, and we will incorporate the related discussion into the revised version and release the code to the community.

---

> ### Author Response · Authors · 2025-08-05
>
> Dear Reviewer ynhN,
>
> We hope this message finds you well. We would like to kindly follow up regarding our recent response to your valuable comments on our submission. Your feedback is very important to us, and we would greatly appreciate any further insights or updates you could provide on the review process.
>
> Thank you very much for your time and consideration. We look forward to hearing from you.
>
> Best regards,
>
> The Authors

---

### Official Review · Reviewer_JUky · 2025-07-03

**Clarity:** 1
**Significance:** 2
**Originality:** 2
**Rating:** 5
**Confidence:** 3

**Summary:**

This paper proposes a framework to generate 3D meshes with parts using diffusion models directly. The model encodes different parts of the objects into different sets of latent vectors, learning an image-conditional diffusion model in the latent space. The diffusion model utilizes local attention within parts and global attention across parts. The paper shows applications in generating part-level meshes with image condition and 3D object-level scene generation.

**Questions:**

1)For training samples that have fewer or more parts than the hyper-parameter N, how do you deal with it?

2)How do you control the number of parts in inference? For example, for a model trained with Z = {z_i} ∈ ℝ^{NK×C}, how do you generate fewer than N parts in inference? What does the shape of Z look like in that case?

3)Is the model able to generate parts inside the objects?

**Ethical Concerns:**

["NO or VERY MINOR ethics concerns only"]

**Final Justification:**

The missing details are added by the authors.

**Limitations:**

yes

**Quality:**

2

**Strengths And Weaknesses:**

Pros:

1)The experiments ablate different components of the framework.

2)The generated shapes have high quality, and the details of the shapes are well preserved.

3)Inference speed is fast.

Cons:

1)There is not enough description about the details of the VAE. What do the encoder and decoder look like?

2)Missing details of the model design, for example,
- what’s the value of C (dimensions of the tokens)?
- How many vertices and faces are generated?
- Are you using a different number of vertices/faces per shape? How is that controlled in inference?

---

> ### Author Rebuttal · Authors · 2025-07-31
>
> Thank you for your feedback. We are encouraged to know that you recognize the high quality of generation results and the fast inference speed of our method. Below, we provide clarifications for the concerns raised. We greatly value your time and effort, and we welcome any follow-up questions or suggestions you may have.
>
> **1. Missing details of the VAE.**
>
> We directly leverage the pretrained TripoSG[1] VAE and do not fine-tune it.
>
> Given a 3D mesh, the VAE first densely samples ~204,800 surface points, each associated with local geometric features such as 3D coordinates and surface normals. From these, it subsamples a smaller set (512 or 2048) to serve as latent queries. Using a set of cross-attention layers, these queries attend to the full dense point set (used as keys and values), allowing the transformer encoder to fuse global shape context into the latent tokens.
>
> The decoder processes these tokens through self-attention, then predicts a truncated signed distance function (TSDF) for query positions via cross-attention. Then, meshes are extracted using Marching Cubes from the decoded TSDF. The VAE uses multi-resolution training, sharing weights across architectures that use varying number of point latents. This enables extrapolation to a higher number of latents at inference time.
>
> The VAE’s loss function includes SDF loss, surface normal loss (using ground-truth normals for fine details), eikonal regularization, and KL regularization.
>
> **2. Missing details of the model design.**
> - The value of $C$ (the token channel dimension).
>
>     Following the TripoSG VAE, we set $C$ to $64$.
>
> - How many vertices and faces are generated?
>
>     The number of vertices and faces generated depends on the resolution used when applying the Marching Cubes algorithm to extract the mesh from the predicted TSDF. Higher resolution settings will result in more vertices and faces, capturing finer geometric details, while lower resolutions will produce fewer, leading to a coarser mesh.
>
> - How do we determine the number of generated vertices and faces?
>
>     The specific number isn't fixed but is determined by the user-specified resolution parameter during the mesh extraction step. This allows flexibility to balance between computational efficiency (fewer vertices/faces) and geometric fidelity (more vertices/faces) based on application needs.
>
> For more details, please refer to the original paper of TripoSG[1].
>
> **3. How to deal with a different number of parts during training?**
>
> We're sorry for the confusion. **The number of parts is not a fixed hyperparameter of our model but rather a variable that can change per datapoint.** Thanks to the use of attention layers, the model can handle varying numbers of parts in the form of a varying number of latent sets in its input. In each training step, the model processes the input object with its actual number of annotated parts, allowing it to learn to handle different configurations. In other words, we train a single model that can handle a varying number of parts per data point both during training and inference.
>
> **4. How do we control the number of parts during inference?**
>
> During inference, the number of parts can be varied by controlling the number of latent sets $N$ in the model. Each latent set corresponds to a part, so adjusting $N$ allows users to specify how many parts they want the model to generate.
>
> **5. Is the model able to generate parts inside the objects?**
>
> **Yes, that model can generate parts inside objects.** For example, in the first example in the "Image to 3D Part-Level Object Generation" section of the supplementary website (a LEGO character), the model generates slot-like structures that are inside the object. These internal components are clearly visible in the provided split-view video, which shows the internal details alongside the external structure, demonstrating the model's capability to generate both exterior and interior parts of objects.
>
>
> We will add all the aforementioned discussions to the revision of this manuscript.
>
> ---
> [1] Li et al., TripoSG: High-Fidelity 3D Shape Synthesis using Large-Scale Rectified Flow Models, arXiv 2025

---

> ### Author Response · Authors · 2025-08-05
>
> Dear Reviewer JUky,
>
> We hope this message finds you well. We would like to kindly follow up regarding our recent response to your valuable comments on our submission. Your feedback is very important to us, and we would greatly appreciate any further insights or updates you could provide on the review process. We have added some supplementary information regarding part number.
>
> 1. How do we control the number of parts during inference?
> We address the issue of part count from two complementary perspectives:
>
> Controllability: Our method enables explicit control over the number of generated parts via the compositional latent space design. This improves controllability by allowing users to directly specify the desired granularity of decomposition. In comparison, existing baselines such as HoloPart[1] and MIDI[2] do not support part count control, thus lacking this level of flexibility. Therefore, our method provides a meaningful contribution by introducing a controllable mechanism that is absent in prior works.
>
> Automatability: To automate part count selection, a vision-language model (VLM), such as GPT-4o/Qwen-VL, can be integrated to perform zero-shot estimation of the number of parts in an image. For instance, given an image of a character, a VLM can infer a typical semantic decomposition (e.g., head, torso, limbs), allowing automatic inference of the appropriate part count. This offers a practical path toward fully automated part-aware generation.
>
> Thank you very much for your time and consideration. We look forward to hearing from you.
>
> Best regards,
>
> The Authors

---

> ### Author Response · Authors · 2025-08-08
>
> Dear Reviewer JUky,
>
> We hope this message finds you well. We would like to kindly follow up regarding our recent response to your valuable comments on our submission. Your feedback is very important to us, and we would greatly appreciate any further insights or updates you could provide on the review process.
>
> Thank you very much for your time and consideration. We look forward to hearing from you.
>
> Best regards,
>
> The Authors

---

### Note · Authors · 2025-08-11

Dear Reviewers, ACs, and SACs,

We sincerely thank you for your suggestions and for recognizing our work’s novelty (**ynhN**, **BRws**, **EYFc**), strong results (**JUky**, **BRws**, **bQqp**, **EYFc**), and thorough experiments (**JUky**, **BRws**, **bQqp**). Your comments have been invaluable in refining our work.

Key concerns before rebuttal:
- **Real-world generation capability.** Our model is trained on synthetic data. To bridge the domain gap, we use a two-stage pipeline: transfer input images to a 3D-rendering style and then perform part-level generation. Experiments on the CO3D dataset confirm its effectiveness.
- **Controllability.**
    - **Number of Parts.** Controlled by setting the latent set count $N$, enabling adjustable granularity. We also discuss how VLMs can automatically estimate and set $N$ based on an input image.
    - **Semantic Control.** We propose a two-stage pipeline that first generates a semantic label map (segmentation map) for the image and then uses this label map to guide the generation. The ABO dataset, which we use for training, includes segmentation map-like images. Therefore, the model works well given the segmentation map.
- **Details.** We’ve provided clarifications on the model architecture, dataset curation, training, and evaluation procedure.

During the rebuttal phase, we've run additional experiments and written detailed responses. We will add these experiments to the manuscript. We're glad that **most of the concerns have been addressed** (**ynhN**, **BRws**, **bQqp**).

Regarding the remaining concerns:
- **Reviewer JUky** called for details on the model design, the adaptation of different part count during training and inference, and the generation ability of the inner structure. **We have provided detailed clarifications.**
- **Reviewer EYFc** raised concerns about the controllability and the editing of non-isolated parts. **We've proposed methods and run experiments to address the controllability concerns**. For editing non-isolated parts, **we proposed an indirect approach to modify the input image to guide the 3D generation.** Since **3D editing is not the primary focus of our work**, we prefer to leave this for future work.

**We promise to release our code, weights, and dataset, and revise our manuscript and project page accordingly.** We hope that the reviewers and the AC can recognize the contributions of this work, and we once again thank you for your efforts.

Best,

The Authors

---

### Decision · Program_Chairs · 2025-09-17

**Decision:**

Accept (poster)

**Comment:**

The paper presents an end-to-end method to generate meshes divided into parts, claimed to be the first of this kind. The idea is to rely on a compositional latent space that represents each 3D part as disentangled latent tokens, and a hierarchical local-global attention mechanism. Authors start from multiple publicly available datasets to curate a 3D dataset with parts annotations.

Before the discussion, the opinions on the paper were particularly borderline; there is a general appreciation for the results and comparison with the state of the art, but I also see there have been several suggestions for further analysis, especially about the number, the controllability, and the generality of the generated parts.

After the discussion, opinions changed more toward acceptance. In summary, Reviewers ynhN, bQqp, and JUky are in favor of acceptance, while Reviewer BRws leans toward it: PartCrafter is recognized as the first end-to-end approach for part-based meshes generated from a single image, and the results have been found to be of high quality.
Instead, Reviewer EYFc , leans toward rejection. Their main concern lies in the controllability of the number of parts, which has to be manually selected by the user. They also have concerns about the editing of the results, though it is not directly the focus of the work.

The AC believes that prompting the user for inputting a part number is a substantial limitation, which might hamper the applicability of the method. The authors suggest this can be inferred by images, but the automatization of this might also not be trivial (e.g., estimation of internal parts), and it is completely left unexplored by the paper. On the other hand, the addressed task is relevant, the experiments provide sufficient evidence about the method's performance and the validation of the individual design components, and it is a fair first step, paving the way for future works. The curated dataset also provides a valuable contribution, instrumental in developing the research field. Hence, the AC leans toward acceptance.